



# Can the boundary profiles at 26N be used to extract buoyancy-forced AMOC signals?

Irene Polo[1,2], Jon Robson[1], Keith Haines[1] and Christopher Thomas[1]

[1]Department of Meteorology, University of Reading, Reading, RG6 6BB, United Kingdom
[2]Departamento de Física de la Tierra y Astrofísica, Universidad Complutense de Madrid, Madrid, 28040, Spain

*Correspondence to*: Irene Polo (ipolo@ucm.es)

**Abstract.** The AMOC circulation is driven both by direct wind stresses and by the buoyancy-driven formation of North Atlantic Deep Water over the Labrador and Nordic Seas. In many models low frequency density variability down the western boundary of the Atlantic basin is linked to changes in the buoyancy forcing over the Atlantic Sub-Polar Gyre (SPG) region, 10  and this is found to explain part of the geostrophic AMOC variability at 26N. In this study, using different experiments with an OGCM, we develop statistical methods to identify characteristic vertical density profiles at 26N at the western and eastern boundaries which relate to the buoyancy-forced AMOC. We show that density anomalies due to anomalous buoyancy forcing over the SPG propagate equatorward along the western Atlantic boundary, through 26N, and then eastward along the equator, and poleward up the eastern Atlantic boundary. The timing of the density anomalies appearing at the eastern and western 15  boundaries at 26N reveals a propagation speed leading to ~2-3 years lags between boundaries with maxima along deeper levels (2600-3000m). Time record required to capture those vertical density profiles in the model is ~26 years. Results suggest that depth structure, and the lagged covariances between the boundaries at 26N, may both provide useful information for detecting density anomalies of high latitude origin in more complex models, and potentially in the observational RAPID array. However, time filtering will be required together with the continuation of the RAPID program in order to extend the time period.

## 1 Introduction

The Atlantic Meridional Overturning Circulation (AMOC) plays a key role in controlling the Earth's energy budget. It transports warm water to the north, overlaying a return flow southward of colder and denser water (Cunningham et al., 2007). Due to its large net heat transport, low frequency variability in this circulation can have an important impact on Atlantic sea surface temperatures and, therefore, on the wider climate (Knight et al., 2005; Sutton and Dong, 2012). Decadal prediction 25  systems have shown that the upper ocean temperatures over the subpolar gyre can be predicted due to the leading role of the ocean heat transport (Robson et al 2012; Hermanson et al 2014; Robson et al., 2017). In order to make these decadal predictions it is essential that we ensure the best ocean initial conditions are available with a well reproduced AMOC.





The RAPID program has been monitoring AMOC and boundary densities at 26N since 2004 (Cunningham et al., 2007;
McCarthy et al., 2015). The observational record so far has revealed a large range of AMOC variability on different time-
scales; from high frequency (Balan Sarojini et al., 2011), to large anomalies persisting at inter-annual time-scales (Blake et al.,
2015; Roberts et al., 2013), or decadal trends (Smeed et al., 2014; Jackson et al 2016). Using different methods, several studies
have investigated the observed weakening (since 2005) of the AMOC and have related the trends to earlier high latitude density
changes (Jackson et al, 2016; Robson et al 2014, 2016).

The AMOC circulation is driven both by direct wind stresses and by the buoyancy-driven formation of North Atlantic Deep
Water (NADW) over the Labrador and Nordic Seas. Theories of the response of the AMOC and the ocean gyres to wind stress
or buoyancy input rely on energy being transmitted through the ocean by planetary Rossby waves, or along the ocean margins
by boundary waves (Johnson and Marshall, 2002; Hirschi et al., 2007; Hodson and Sutton, 2012; Jackson et al., 2016). In
particular, changes in the NADW may produce a chain of events in the North Atlantic on a range of time-scales from months
to decades. The adjustment has been studied in an extensive literature. Some model studies (Kawase, 1987; Huang et al., 2000;
Johnson and Marshall, 2002; Getzlaff et al 2005; Marshall and Johnson, 2013) suggest that AMOC anomalies propagate with
boundary Kelvin wave speeds resulting in a very short lead time (of order a few months) between subpolar and subtropical
AMOC changes. Roussenov et al (2008) suggested that this boundary propagation may also involve higher mode Kelvin and
topographic Rossby waves, leading to longer propagation times (of order years). The advection of the NADW outflow also
moves down the western boundary more slowly in the Deep Western Boundary Current (DWBC), although lagrangian float
observations show that a large fraction of this NADW moves away from the boundary and enters the ocean interior near the
Flemish Cap and the Grand Banks (Bower et al., 2009). Using a coupled climate model Zhang (2010) and Zhang et al. (2011)
showed AMOC variations associated with NADW formation propagating more in line with the advection speed, with much
longer lead times (several years) between subpolar and subtropical AMOC variations. Getzlaff et al. (2005) have shown that
the high latitude adjustment to AMOC anomalies can result from a superposition of a fast wave response and a slower advective
signal in ocean model experiments with different resolutions. Interestingly, the speed of propagation along boundaries of the
density/velocity anomalies related to AMOC changes is found to be model-resolution dependent (Getzlaff et al 2005; Hodson
and Sutton, 2012). These propagating density anomalies will also affect the geostrophic AMOC variability at 26N.


Model simulations clearly show a large range of mechanisms leading to AMOC variability at latitude 26N (Hirschi et al., 2007;
Biastoch et al., 2008; Cabanes et al., 2008; Duchez et al., 2011; Polo et al., 2014; Pillar et al., 2016). Buoyancy forcing generally
operates from inter-annual to decadal time-scales, while the wind forcing mostly acts from intra-seasonal to inter-annual time-
scales (Cabanes, et al., 2008; Kanzow et al., 2010; Duchez et al., 2011; Polo et al., 2014; Pillar et al., 2016). Using an adjoint
OGCM, Pillar et al (2016) have found that inter-annual to inter-decadal AMOC variability of ~5 Sv amplitude can be excited
by heat fluxes in the subpolar North Atlantic, with freshwater fluxes playing a more minor role. Due to the adjustment from
higher latitudes, Western Boundary (WB) density anomalies explain most of the variance in the zonal density gradients, and





hence geostrophic transports, at 26N, especially at decadal time-scales (Hirst et al., 2007; Polo et al., 2014). The Eastern Boundary (EB) explains only a small part of the inter-annual variability in zonal density gradients in the upper 1500m, and
this is mostly due to local wind forcing (Polo et al., 2014).

Despite the many studies showing boundary wave connections between the Labrador Sea and lower latitudes, and their importance for the AMOC, less work has been done on the vertical structure of these anomalies, and yet it is the vertical density structure at 26N that is primarily measured by the RAPID array. We now benefit from more than 10 years of boundary
density records at 26N, and therefore can consider how best to use the vertical structure in these data to study the lower frequency variability. If low frequency signals can be identified from the vertical structure this would help us to assimilate the most important signals that need to be reproduced in climate forecast models. This poses the question we address in this paper; *can we extract the buoyancy forced signals from vertical density profiles, such as those sampled at 26N?*

An earlier attempt to extract buoyancy signals at 26N was made by Polo et al (2014, hereafter PA14) using a NEMO 1° OGCM forced with full ECMWF reanalysis meteorology from 1958-2010, denoted as the CTRL experiment. This CTRL experiment was compared to runs using only inter-annual wind or buoyancy forcing, allowing separation of buoyancy from wind forced variations in the AMOC. PA14 found that the buoyancy-forced AMOC anomalies at 26N could be related to changes in deep water formation over the Labrador Sea some years before. Although they showed a coherent WB vertical signal at 26N, they
were not successful in isolating the buoyancy forced signal in the CTRL experiment, due to the confounding influence of the wind forced variability. They did not look in detail at the propagation or how the vertical structure associated with the buoyancy forced anomalies develops. In the present work we extend the work of PA14 by (i) developing statistical means of isolating the buoyancy forced AMOC variability from the full variability in the CTRL using the density profiles at 26N; (ii) analysing the propagation of the buoyancy forced signals from the Labrador Sea down to the subtropics; and (iii) developing statistical
covariance relationships linking the AMOC to the Labrador Sea that might potentially be used in a data assimilation context to modify the low frequency AMOC variability. The diagnostics developed are also tested on RAPID observations and on output from the state-of-the-art coupled model HadGEM3-GC2 (Williams et al 2015), which has a ¼° NEMO global ocean.

In this paper Section 2 presents the methodology used to analyse AMOC variability and its sources in several runs of the
NEMO model (1°x1° horizontal resolution) driven by different components of ECMWF atmospheric forcing from 1960-2012, as in PA14, but now including some validation of the boundary density variability in the CTRL run against the RAPID observations. Sections 3 and 4 describe the modes of the 26N density profile variability in the model and the associated propagation occurring upstream and downstream, respectively. Section 5 describes a statistical analysis of boundary densities in the RAPID observations and compares these modes with the NEMO experiments. Section 6 discusses density variability in





the coupled experiments GC2, Section 7 describes the limitations of the interpretations, as well as possible applications e.g. in data assimilation. Finally, Section 8 summarizes the main conclusions.

## 2 Methodology

This section describes the model experiments and statistical methods used to understand the boundary density variability and its relation to the AMOC variability.

**2.1 Forced experiments**

The forced ocean-only model (hereafter NEMO1) is based on NEMO V3.0; it uses the tripolar ORCA grid in a global configuration with 1°x1° horizontal resolution and a tropical meridional refinement to 1/3°. The model has 42 vertical levels with thicknesses ranging from 10m at the surface to 250m at the ocean bottom. Initial conditions are taken from the second iteration of a 50-yr cyclic model spin up, each cycle spanning the period 1958–2009 (Balmaseda et al., 2013). The model is

forced with daily atmospheric fluxes as boundary conditions taken from the ECMWF Re-Analysis ERA-40; (Uppala et al. 2005) from 1958 to 1978, and the Interim ECMWF Re-Analysis (ERA-Interim; Dee et al. 2011) from 1979 to 2009.

The control experiment (CTRL) is forced with time-varying daily surface heat, freshwater and momentum fluxes for the period 1958–2009. The sea surface temperature (SST) is weakly relaxed to daily values with a relaxation time scale of ~1 month, while the sea surface salinity (SSS) is restored to climatological SSS with a time scale of 1 year. There is no ice model; instead,

wherever the sea ice concentration in the observations exceeds 55%, the model SSTs are nudged more strongly (1-day time scale) to the freezing point (-1.88°C). The restoration to SSS and SST is stronger under sea ice (30 days and 1 day, respectively).

Following the work of PA14 we also have a set of simulations where the momentum and buoyancy forcing is decoupled from

one-another. In the experiment referred to as BUOY, the momentum flux is taken from the ERA-Interim 1989–2009 seasonal climatology, while the buoyancy forcing (heat, freshwater flux, and SST) is still inter-annually varying. In the experiment referred to as WIND, the momentum flux is fully varying, but the buoyancy forcing is from the same seasonal climatology. These experiments allow us to identify and distinguish the AMOC signals and processes associated with buoyancy and wind forcing, to the extent that they are independent. We use the BUOY experiment as reference for the buoyancy-forced only

signals, and the CTRL experiment as the "truth" which includes both buoyancy and wind forcings, as well as the interaction between them. Where appropriate, we also include the WIND experiment and SUM (as the sum of anomalies from BUOY and WIND). Results are discussed in Sections 3, 4.



## 2.2 Coupled experiment

We also analyse 120 years of monthly-mean data from a control run of the high-resolution coupled ocean-atmosphere model HadGEM3-GC2 (hereafter GC2, Williams et al 2015). The ocean component is NEMO v3.4 with the ORCA025 tripolar grid configuration, using Met Office parameters for "Global Ocean 5.0" (GO5.0, Megann et al 2014), with the CICE sea-ice model. The atmosphere component is GA 6.0 of the Met Office Unified Model (UM; Walters et al 2011) at a resolution of N216 (~60Km in mid-latitudes) and 85 levels. This model was used in the Met Office seasonal and decadal prediction systems

(GloSea5 and DePreSys3 respectively). The model has been used to study the North Atlantic variability and its predictability (Menary et al 2015; Williams et al 2015; Ortega et al 2017; Robson et al., 2016). Results are shown in Section 6.

## 2.3 Model evaluation

We use the RAPID array (McCarthy et al., 2015; Smeed et al 2017) to evaluate the boundary densities in the model. We use the merged profiles at the Western Boundary (26.52N, 76.74W) and Eastern Boundary (26.99N, 16.23W) for the period April

2004 to February 2017.

NEMO1 and GC2 are both able to capture important aspects of the observed boundary density profiles such as the mean vertical density gradients ($N^2$, Fig. S1a). On the WB the profiles are similar between 1500m to 4500m but the model stratification is stronger between 300-700m. The EB profiles are similar at all levels below 500m (Fig. S1a). However, the

NEMO1 model underestimates the density variance at all levels, especially at the WB, while GC2 has a more realistic variance on the WB at depth (Fig. S1b).

The AMOC at 26N in NEMO1 has a time mean (12Sv) and maximum (18 Sv) at a depth of 1000m in the CTRL experiment. The mean AMOC is higher in the BUOY experiment by ~2 Sv. The AMOC measured at 1000m has a prominent trend in the

BUOY experiment (+3.2 Sv in 52 years), but the trend is not significant for the CTRL (-0.2 Sv in 52 years). The AMOC seasonal cycle (not shown) in the CTRL presents a maximum in boreal winter with a secondary peak in boreal summer, which is also reproduced in the BUOY experiment. The annual cycle defined as the difference between the maximum (in boreal winter) and the minimum (in boreal spring) is 3.9 and 4.9 Sv for the CTRL and BUOY experiments. After removing the linear trend and the seasonal cycle the standard deviation of both experiments is similar: 1.89 and 1.43 Sv for CTRL and BUOY

respectively (see also Fig. S2 for the AMOC distributions). The AMOC at 26N in the RAPID observations presents a mean and maximum of 17 and 31 Sv respectively from 2004-2014, with monthly standard deviation of 4.35 Sv. Trends have been reported for RAPID data of -0.6Sv/year (Smeed et al 2014; 2018) which could be part of a longer variation cycle (Smeed et al 2018, Jackson et al 2016). Results from the modes of variability at the western boundary density profile are shown in section 5.





### 2.4 Statistical analysis

Model experiments are first sampled at the western and eastern Atlantic boundaries at 26N to simulate the monthly-mean density profiles from the RAPID array. Empirical Orthogonal Function (EOF) analysis of these density profiles is used to obtain vertical modes of density variability and related timeseries of Principal Components (PCs) which together represent the largest fractions of the total variance (Bretherton et al., 1992).

Before calculating the EOFs, the data are processed to remove the seasonal cycle and linear trends. Unlike in PA14, density anomalies are weighted by the thickness of each layer to ensure that all points are appropriately represented for the total density variability. The EOF analysis is computed for the individual boundaries; Western Boundary (WB) and Eastern Boundary (EB), and also for the combined anomalies at both boundaries. Finally, we have explored the combined EOFs by time lagging the eastern boundary variability in order to understand related signals from both boundaries.

Regression analysis of the PC time series associated with these EOFs on other fields (e.g. 3D density) allows us to detect spatial patterns and depth structures of the propagating modes associated with the EOFs at 26N. We show regression and the correlation coefficients where they are statistically significant at the 95% confidence level, according to a Student's t-test for the effective number of degrees of freedom (Metz, 1991).

### 2.5 Spectral analysis

In order to remove the high frequencies in the timeseries in section 3, we have used a one-year running mean filter. This filtered timeseries is obtained by taking the average of a data subset (13 months) which is centered in a monthly time step (von Storch and Zwiers, 1999).

Spectral analysis is used to decompose timeseries to show signals that lie within different frequency bands. The analysis is performed in order to identify the frequencies involved in the propagation of density anomalies at different depths. Power spectra of the time series are obtained using the multi-taper method, which provides more degrees of freedom and therefore more significance (Thomson 1982). The power spectra are tested against the hypothesis that the signals are generated by a first-order autoregressive process AR(1) with the same time-scale as the original, yielding a red noise spectrum, and the 95% confidence limit for the rejection of the red noise hypothesis is applied. Additionally, when we have a large internal variability, we use the decomposition in order to filter some of the timeseries using a Lanczos (1956) filter. This is done in particular for the control GC2 run simulation.

We have used the Radon Transform (RT) function (Dean 1983) in order to estimate the phase speed of propagation of density anomalies. The angle of the maximum RT standard deviation determines the propagation phase speed. We calculate the RT





every 0.1 degrees. The phase speed averaged for the Hovmöller diagrams has been estimated at 3000m level for CTRL and BUOY experiments.

## 3 Modes of vertical density variability at the 26N boundaries

**3.1 Linearity of AMOC and the boundary density signals in the NEMO1 model**

The forced ocean model experiments enable us to isolate the boundary density variability associated with buoyancy forcing i.e. from the BUOY experiment, as noted by PA14. We focus here on developing the vertical density fingerprint of this signal and using it to identify the buoyancy forced AMOC signal as it appears in the CTRL run.

Figure 1a shows the monthly AMOC (defined as total AMOC minus Ekman component) variability, defined as the integral of the meridional transport at 26N down to 1028m, for both the CTRL and BUOY experiments (as in Fig. 1a in PA14). There is a prominent decadal signal in both CTRL and BUOY with peaks in 1975, 1985 and 1995, although CTRL also shows additional monthly and inter-annual variability. The monthly-mean timeseries correlate at 0.43, but the correlation rises to 0.62 when using a 1 year running mean filter (Fig. 1b). Wind forced inter-annual variability explains most of the remaining differences;

when the 1-year smoothed AMOC anomalies from BUOY and WIND are summed the correlation with CTRL rises to 0.86 (SUM in Fig 1b).

The majority of the boundary density variability is also recreated in BUOY and WIND. The correlations between boundary density anomalies in the CTRL and SUM are shown as a function of depth in Fig. 1c-d. For the WB, most of the variability is

linearly reproduced by SUM from 1800m to 4000m (Fig. 1c and Fig. S3). For the EB SUM explains most of the variability seen in the CTRL experiment at all depths (Fig. 1d and Fig. S3). A 1yr low pass filtering does not influence the correlations for the WB, although for the EB filtering reduces the correlation at some depths. We now relate this density variability with the AMOC signals in Fig 1a, b.

### 3.2 EOFs of boundary density profiles

Figure 2 shows the Principal Component time series of the first EOF computed using monthly density profiles on the western and eastern boundary at 26N for the CTRL (blue) and BUOY (red) experiments. As density profiles near the surface contain significant noise, we calculate the PCs for both full depth profiles (f0m, Fig. 2a-b) and only from 800m downwards (f800m, Fig. 2c-d). The full depth PCs in Fig. 2a both show substantial high frequency noise, and the CTRL timeseries does not correlate well with AMOC variability from Fig 1a. However, when only retaining densities below 800m, in Fig. 2c, the BUOY and

CTRL PCs closely match each other (r=0.80) and both now correlate with the AMOC time series in Fig. 1a (r=0.33 and r=0.88 for the CTRL and BUOY respectively). These f800m EOFs also explain more of the deeper density variance. Note that, adding





a 1-yr filter to the PC-CTRL timeseries increases the correlations with the CTRL AMOC variability to 0.47 (Fig. 2c). However, for the BUOY experiment the temporal filter has little impact on the correlation between PC and AMOC timeseries (0.89).

Figures 2b and d show the corresponding eastern boundary density EOF timeseries for full depth and below 800m variability. The full depth PC is rather different to that on the western boundary, and also to the AMOC variability itself. However, when only the deep variability is retained, the inter-annual variability in BUOY is more similar to the western boundary buoyancy driven variability. In CTRL there is still considerable high frequency wind generated variability, however, when a 1-yr filter is introduced the buoyancy forced AMOC related signal also becomes visible on the eastern boundary, and the correlation with
the AMOC rises to 0.42 (Fig. 2d).  All correlations are summarised in Table 1.

In order to understand the gain of truncating the density profiles in the EOFs and theirs limits, Fig. 3a-b summarises correlations between leading PCs at the boundaries and AMOC timeseries in the CTRL experiment by increasing the truncation level for the density profiles. For the WB, maximum correlation is found simultaneously at all truncations especially for deeper levels
(Fig. 3a). In contrast for the EB simultaneous correlations are always low even with a 1 year filter but correlations increase greatly when a lag to the AMOC is applied (Fig. 3b). Correlations still require a 1 year filter to remove noise but now peak at 0.7 with a lag around 2-3 years, with the deeper signals also showing the longer lags. Similar lag increases of EB correlation is seen for the BUOY experiment (not shown). The nature of this lag in the EB-AMOC correlation is related to the link between boundaries and it will be discussed along in the next section.


Figure 3c-d shows the vertical structure of density anomalies associated with the leading EOF modes for both boundaries using the 800m depth truncation and monthly-mean data for the three experiments CTRL, BUOY and WIND. The leading EOFs are very similar between CTRL and BUOY on the WB, with maxima between 1200m and 4000m. Note that in PA14 Fig. 3b, their PC with only 500m depth truncation was substantially different and was mainly wind-driven. The WIND experiment has much
less variability at greater depths and the PC is uncorrelated with the PC-WB from CTRL (r=0.09, Fig. 3c).  Therefore on the WB, Fig.s 2c and 3c show that the EOF analysis successfully extracts the buoyancy-forced signal related to the AMOC in the CTRL.

On the EB the leading EOFs, even below 800m, show more correspondence between CTRL and WIND (Fig. 3d, with PC
correlations r=0.87). This explains why further filtering and the application of a lag to the PC timeseries in Fig 2d, 3b is needed to extract the weaker buoyancy-forced AMOC-related signal on the EB. The relationship between these buoyancy driven density variations at both boundaries is now explored further.


### 3.3 Relationship between boundaries

Figure 4a shows lead-lagged correlations between the WB leading PCs (computed from 800m depth) and the EB at different
depth truncations in the CTRL experiment. Dashed (solid) lines indicate PCs without (with) the application of 1year running
mean filtering. For truncations deeper than 1600m the highest correlation is found when EB is lagged, up to 30 months (Fig.
4a), revealing longer links between boundaries for deeper levels. However this lag is much less clear when shallower depth
are retained and substantial WB-EB density correlations can still be seen with lag0, unlike in Fig 3b. Even in the BUOY
experiment only when the EB EOFs are truncated to below 1600m is a strong lag clearly seen between the boundaries, again
reaching up to 30 months for the deepest signals (Fig. 4b).

In order to reduce upper level noise at the EB and to bring out the deep density signal connecting the boundaries more clearly
we compute the combined EOF while truncating the EB to below 1600m, which shows a maximum in the WB-EB correlations
for CTRL (Fig. 4a). The new combined EOF shows similar WB structure for all lags (Fig. 4c) with a deeper signal around
2000m on the EB (Fig. 4d). Figure 4e shows the timeseries, which is very similar to the PC1-WB in Fig. 2c. The new combined
EOF explains more variance in CTRL (43% compared to 40%) and a slighly longer lag between WB and EB (18 months in
CTRL and 25 months in BUOY, not shown).

We conclude that the deep densities on the EB contain a very clear signal of the buoyancy forced AMOC variability but that
this signal plays no detectable role in the direct (lag 0) control of the AMOC. The EB lag can also be seen in relation to PC-
WB density although the signal is less clearly lagged, probably reflecting the noise still present in the upper layer densities on
the WB.

The WB clearly contains the core density information on the buoyancy-forced AMOC changes at low frequencies. Using PC1-
WB, we will now identify the propagating density signal connecting the boundaries (section 4) and search for similar signals
in RAPID data (section 5), and in the higher resolution GC2 model (section 6).

### 4 Propagation of the buoyancy-driven signals

Motivated by the lagged signal at the EB, we analyse the i) spatial coherence of the anomalies at deeper levels (~3000m), and
ii) the propagation fingerprints of the connecting signals.

### 4.1 Spatial regression patterns

Figure 5 shows the spatial density anomalies averaged 2700-3000m, regressed onto the PC1-WB for the CTRL and BUOY
experiments respectively, at different lags. At these deep levels the regression patterns in CTRL (Fig. 5a-c) and BUOY (Fig.
5d-f) are very similar to each other. This agreement suggests that, although the magnitude of the regressions are stronger in


BUOY, we are identifying the same signal in both experiments. The density signal in the Labrador Sea appears from lag -30
(Fig. 5a, d) and intensifies and propagates down the WB to the Equator (Fig. 5b, e), and then across the equator and poleward
at the EB (Fig. 5c,f).

Density regressions at shallower levels (900-1300m) against PC1-WB in the CTRL experiment (suggested by the maximum
in the EOF profiles in Fig. 3) also finds anomalies beginning near the Labrador Sea, leading PC1-WB by 30 months, and a
pattern of equatorial Kelvin and Rossby-waves as in Johnson and Marshall (2002) from lag -30 to lag ~0 (Fig. S5). However,
the absence of such tropical signals in BUOY at long lags (lag -30) shows that these could be wind driven Ekman pumping
signals (see supplementary material in Fig. S6). Therefore, we concentrate on the deeper signal which is clearly related to
buoyancy.

**4.2 Wave path and phase speed**

Figure 6a shows the wave-track defined following the signals in Fig. 5 and using the topography at the 3269 m model level.
The wave-track starts in the Labrador Sea (60N) and proceeds southwards to the equator. We plot it along the equator and then
North along the eastern boundary to 55N. The path avoids entering into the Gulf of Mexico but at these depths this is not
expected.

The Hovmöller diagram along this path (Fig. 6b) shows the propagation related to peaks in the deep water formation at high
latitudes for the CTRL experiment (BUOY is very similar, not shown). Density anomalies propagate continuously along the
track from the Labrador Sea around to the British Isles. The propagation shows density maxima in 1975, 1985 and 1995, also
seen as peak-AMOC years in Figs 1-2. Additionally, the propagation speed from the Radon transform is similar for both
experiments (0.41 and 0.31m/s for the BUOY and CTRL respectively). This phase speed is consistent with the lags found
between boundaries (i.e. density anomalies in CTRL will take ~25 months to travel between WB and EB following the defined
track). Background currents at 2700-3000m level for the CTRL is shown in supplementary material (Fig. S4).

Figure 7 shows the regression of density anomalies from Fig. 6b onto PC1-WB for the CTRL and BUOY experiments. The
density anomalies in both experiments show a continuously propagating pattern from the Labrador Sea right around to 40N on
the eastern boundary. The signal is stronger in BUOY, as noted in Fig. 5, but otherwise the regression patterns are very similar.

Importantly these propagating buoyancy-related signals are clearly seen in the CTRL experiment where wind and buoyancy
forcing are both applied, again suggesting that the analysis is extracting the same buoyancy forced processes. Therefore, the
diagnostic methods developed should allow identification of similar signals in other models and in the observations. In the
next section we look for similar buoyancy-forced signals in the RAPID observations.





# 5 Modes of WB density variability in RAPID data

The RAPID timeseries dataset is considerably shorter than the datasets analysed for buoyancy signals in the NEMO model. Nevertheless Fig. 8 uses the same EOF analysis on the WB density profile for the RAPID array and for the NEMO experiments for the common period 2004-2010. The first two leading EOFs and their PCs indicate inter-annual variability in RAPID data

(black lines in Fig. 8a, c). EOF1 has a maximum density anomaly at ~1000m (with a maximum of 0.04 kg/m3 in Fig. 8b), while the second mode describes variations at deeper levels (~3000m, Fig. 8d).

The density EOFs of the WB in both the CTRL (blue lines) and WIND (green lines) experiments for the common period 2004-2010 look quite similar to the EOFs from the RAPID array, in particular with EOF1 peaking at around 1000m and EOF2

peaking much deeper at ~3000m (Fig. 8b, d). However in BUOY (red lines) the absence of inter-annual wind variability means that the deep 3000m peak shows up as EOF1 which is more comparable to EOF2 from RAPID, while BUOY EOF2 shows very little clear density signal.

The PC1 time series in CTRL and WIND also look very similar to PC1 from RAPID, reaching a peak in 2007 and declining

to 2010 (Fig. 8a). For these common 6 years of simulation 2004-2010, the wind-forced inter-annual density variability on the WB is remarkably well captured by the model.

Figure 9 shows the statistics of the correlation between the leading WB mode for CTRL and both, WIND (Fig. 9a) and BUOY (Fig. 9b) experiments using different numbers of years for each sub-period (x-axis). Figure 9a (first box) shows the test of

sampling 6 years periods (as we have in 6 years in the common period in Fig. 8a). For all 6 year periods selected between 1958 and 2010 we also found that EOF1 and PC1 in CTRL and WIND agreed well (Fig. 9a, first box), even in periods when the buoyancy signal was known to be changing rapidly. This dominance of the wind forcing over short time periods is not surprising and was noted previously (PA14).

We find that typically 16 years of data are needed to find a significant (above 0.35) correlation between CTRL and BUOY (Fig. 9b), however the leading mode in CTRL may still be a mix of wind and buoyancy forcing (as seen from Fig. 9a). The best extraction of buoyancy forcing signals occurs when we have periods longer than 35 years, when the wind-forced signal nearly disappears (Fig. 9a). Figure 9c shows the correlation between the PC1-WB and the AMOC timeseries for the CTRL and different periods. This would represent how much variance of AMOC can be explained by the PC1-WB. For periods with

more than 25 years, PC1-WB in CTRL experiment is able to explain more than 25% of the AMOC variance (r>0.5), therefore, it is still able to extract an AMOC-related signal at 26N (Fig. 9c).

Although EOF2 from CTRL and RAPID both represent deeper density variability (Fig. 8d) they are still dominated by wind forcing over this short time period, and we note that EOF2 from WIND shows the same deep density peak. The PC2s from

CTRL and WIND show very similar time series and even the RAPID PC2 shows a considerable level of agreement with CTRL and WIND (Fig. 8c). However PC1 from BUOY, which has a similar EOF but represents only the buoyancy forced component, has only lower-frequency changes with no relationship to the other timeseries (Fig. 8a). The BUOY PC2 timeseries (Fig. 8c) also shows no comparable variability.

Therefore, the short record of the RAPID array would not allow us to follow buoyancy-forced signal from the NEMO1 model. Hence, it appears likely that the variability seen here in the RAPID record, both at shallow and deeper depths, is mainly related to wind-forcing (both PC1 and PC2 in CTRL experiment and RAPID show agreement in Fig. 8). Note that the same EOF analysis using the longer record now available for RAPID data (2004-2018), but not for these model results, still gives similar density profiles to Fig. 8 (not shown).


The NEMO1 model results suggest that the lower frequency buoyancy forced signals from higher latitudes may start to dominate over the wind forced signals after ~25 years of RAPID data have been collected, when their leading density variability should show up at deeper depths ~3000m. In the next section we look at the ability of the analysis to extract buoyancy forced signals and their propagation in higher resolution HadGEM3-GC2 coupled model run, which is the current UK operational

coupled model.

## 6 Boundary density in a high resolution coupled model

### 6.1 Density propagation in GC2

Figure 10a shows a wave track for the GC2 model (section 2.2) bathymetry using the 3138m level boundary, and figure 10b shows the Hovmöller diagram of the density anomalies for 120 years along this track , after applying a 2 year low pass filter

as GC2 is much noisier at higher frequencies (see methods). We notice that the high frequency density anomalies (<2years period) in a similar Hovmöller are not presenting propagating signals but very noisy signals (not shown), therefore we suggest that the high frequency signal is dominated by local Ekman pumping.

Anomalies propagate down the WB from ~40N (track point 200) to the equator and across to the EB. These signals can be

traced back across the Gulf Stream to subpolar latitudes (points 100-200), but only appear as lower frequency decadal variations in the subpolar gyre and into the Labrador Sea (points 0-100). The Radon Transform phase speed is ~2m/s, which is faster than the phase speed calculated in NEMO1 at the same depth (Fig. 6), and closer to the theoretical and observed Kelvin wave propagation speed (Polo et al 2008). Therefore, as the density anomalies propagate down the deep western boundary, we would expect to find this deep density variability signal using the EOF analysis.






## 6.2 EOFs at the WB in GC2

Figure 11a shows the vertical density profiles associated with the first two EOFs from 800m downwards at the 26N WB in the GC2 experiment. The profile location is the closest grid-point to the western wall at the Bahamas, which reaches the bottom at 3200m, and the first 2 EOFs explain more than 70% of the total variance. The EOF1 shows an equivalent barotropic vertical

structure peaking near the bottom ~3000m (blue line) while EOF2 changes sign between 900m and 3000m (red line).

The PC1 timeseries associated with EOF1 is plotted in Fig. 11b (blue line). Unlike in NEMO1 this PC1 shows high frequency variability which is nevertheless still correlated with the AMOC-Ekman (i.e. the peak AMOC stream function at 900m after the variability due to Ekman has been removed) without filtering at r=0.45, rising to r=0.49 with high-pass (<2 years) filtering.

However, PC1 becomes less correlated with the AMOC after 2 year low-pass filtering, r=0.25 (Table 2). In contrast, for PC2 the unfiltered correlation with the AMOC-Ekman is low (r=0.13) but this increases with 2 years low pass filtering (r=0.32, Fig. 11c, Table 2). As density anomalies at deep levels are able to be excited by wind alone (already seen in both NEMO1 and the RAPID observations in Fig. 8), the two EOFs in GC2 both capture some wind and buoyancy forcing, with variability signals in both PC1 and PC2.


Although the PC1 and PC2 are orthogonal by construction and thus correlation between timeseries is zero, after time filtering the PCs timeseries, the modes are correlated (r= 0.41) and the lead-lag correlations with AMOC present a cycle between vertical profiles modes, with a 16 months between the peaks (Fig. 11d). This indicates the limitations of extracting low frequency AMOC-related signals in complex environments using linear methods.


High frequency PC1-WB (<2years) represents high frequency density signal at 26N, which could be wind-forced. It is correlated with the AMOC and is independent of the number of years used to identify it (Fig. 11e). Low frequency PC1-WB (>2years) represent a small part of the low frequency density signal at 26N (r<0.2) and it is independent of the number of years (not shown). PC2-WB correlates better with a lower-frequency AMOC signal (>2years) and is also independent of the number

of years used to identify it (Fig. 11f).

If we filter the density anomalies prior to performing the EOF, then the leading mode corresponds to PC2-WB seen here. This confirms that we cannot isolate the low frequencies by identifying a deep density signature as works well in NEMO1, therefore time-filtering is needed to identify the inter-annual buoyancy signal in GC2 as a leading mode. Nevertheless, the buoyancy

signal is still traceable emerging from the Labrador Sea, and is well captured in PCs, representing relevant information on AMOC variability.





Here we summarise the comparison between density profile modes in different environments as follows:

i) In the NEMO1 CTRL simulation with 52 years of data, we are able to find a vertical density pattern at 26N that is buoyancy-forced (i.e. similar to the mode in the BUOY experiment). This has maximum density anomalies at 1500-3000m and corresponds to low-frequency variability of the AMOC. The source is the density changes over the Labrador Sea. In comparison, the leading density mode for the WIND experiment has a maximum at 1000m and represents inter-annual variability of the AMOC.


ii) However, if the same methodology is applied over shorter periods, the wind-forced variability dominates both the first 2 EOFs with density anomalies at 1000m and 3000m. The limit for the time-period is about 25 years to extract buoyancy-forced signals that can be related to the AMOC at lower frequencies.

iii) When CTRL is compared with the RAPID array PCs, we find similar vertical profiles in the CTRL and observed PCs. Suggesting the short period of RAPID does not allow to extract relevant buoyancy-signals. Wind-forced signals are predominant showing density anomalies that can be also relevant for the geostrophic part of the AMOC at 1000m (but without any lag and at inter-annual scales).

iv) The same methodology applied to a more complex GC2 earth system model results in a leading PC that shows positive anomalies between 1000 and 3000m. This PC1 is related to the AMOC at short inter-annual timescales (predominantly wind-forced). The PC2 mode shows reversing density anomalies between 1000 and 3000m and is related to AMOC at lower frequencies (period >2 years). The analysis shows that this PC will be dominant independently from the number of years used in the calculation, although will have less spread for periods longer than ~26 years (Fig. 11e-f).


The EOF below 800m method seems to be appropriated to detect buoyancy forced signals from density profiles if we have more than 25 years of data in NEMO1. However, a Lanzcos time filtering for periods > 2years prior to the EOF analysis is recommended in a more complex environments as required for RAPID observations and coupled models. From similarities between NEMO, RAPID and GC2, we conclude that the density profile mode that is most likely buoyancy-forced corresponds

to density anomalies at deep levels (3000m) that covary negatively with density anomalies at upper levels (1000m). Timeseries should be filtered with periods >2years and PC correlates with AMOC at 26N 1000m at same frequencies.



## 7 Discussion

In this work we have used model output and statistical methods to identify vertical density profiles along the boundaries that
are consistent with the buoyancy-forced variability. We have shown that the most relevant profile at 26N is found at the WB
using EOF analysis after truncating the density profile from 800m (PC1-WB). This truncation is very effective in emphasising
low frequency (decadal time-scales) signals and in NEMO1 negates the need for temporal filtering, which can also add spurious
signals or lead to excess smoothing. Caveats that warrant further discussion include the differences between the EOFs in
NEMO1 and GC2, and the role of the EB.

We note that vertical profile of the WB EOFs in GC2 and NEMO1 are different, especially in the top 1500m, and the
frequencies of the dominant variability in boundary signals in GC2 are higher than the decadal signature seen in NEMO1. We
noticed that shallow (<1500m) density signal related to PC1-WB in NEMO have different timing in CTRL and BUOY
experiments at the tropics (Fig. S5-6), suggesting that at 26N wind forcing is modulating the buoyancy-forced density signal
in CTRL. This is also an argument to suggest that in GC2 the shallow signal is more probably wind-forced signal.

Although PCs GC2 contains the low-frequency AMOC-related variance it is perhaps not surprising that the details of the WB
EOFs are different between GC2 and NEMO1 given the range of AMOC variability in models (Biastoch et al., 2008; Cabanes
et al., 2008; PA14; Ortega et al., 2017). The 1° horizontal resolution, and even the ¼° model, may still be too coarse to correctly
capture propagating boundary signals (Johnson and Marshall, 2002; Getzlaff et al., 2005; Hodson and Sutton, 2012) from the
Labrador Sea. Therefore, we may not expect the exact details of the boundary density EOFs, or the phase speeds nor the phase
lags identified from the boundary and Labrador Sea signals to be very realistic. Nevertheless, the methods reveal, in two very
different modelling environments, boundary signals consistently related to the geostrophic AMOC at 26N simultaneously.

It is worth notice that time filtering is needed to see a clear signal due to noise from high frequency wind-forcing in GC2.
Wind forcing can be projected onto density anomalies at deep levels (as it is seen in RAPID data and GC2), it is possible that
in the observations similar time-filtering would be needed in order to extract the buoyancy-forced signals. Understanding the
differences between these boundary density EOFs between models, would be useful to interpret observations as the record
becomes longer, and it should be a focus of further work.

In Section 3.3, we tested the value of using WB and EB together in a single EOF. The fact that the combining the WB and EB
at different lags in a singular structure does not improve the explained variance of the EOF WB alone reveals that the WB
contains most of the variability at decadal time-scales. This is in agreement with previous results by PA14 and also with the
propagating signal from the Labrador Sea that can reach the subtropics along the WB (as in shown by Jackson et al., 2016),
but it is not that obvious for the EB. However, we must be clear that we are not saying that EB observations do not make up



an important component of the RAPID array observations. Indeed, it has already been shown in observations (Kanzow et al, 2010; Duchez et al 2011), and in models (PA14), that the EB is important for understanding the wind-forced variability in the observed AMOC at 26N from sub-annual to inter-annual time-scales. Indeed, in our own study the use of the EB was important to isolate the density propagation at 3000m depth which is an AMOC-related signature. Therefore, the EB observations are
still important in understanding the role of decadal-buoyancy forced variability.

We have found density variability in RAPID which the models suggest could be buoyancy-forced at high latitudes (PC1-WB in NEMO1). However, the temporal variability suggests that wind forcing is still dominating at these short timescales. We have to wait for more years of RAPID data, which could allow the buoyancy forced variance to dominate this mode or give
this mode EOF1 status in the decomposition. Although it may be helpful to do some time filtering which could allow buoyancy forced signal to be found in shorter periods of data, the continuation of the RAPID array would be crucial in order to understand the wind-forced inter-annual variability and also the link between the subpolar North Atlantic and the AMOC at 26N.

## 8 Conclusions

In this work we have used NEMO1 OGCM experiments which separate buoyancy and wind forced signals (BUOY and WIND
experiments; Polo et al., 2014), together with statistical techniques, to develop methods to extract the Atlantic boundary density profile signatures at 26N most associated with the buoyancy-forced AMOC signals from an experiment with both buoyancy and wind forced variability (CTRL). After finding the "best" vertical profile on the western boundary, we describe the temporal-spatial structures related to this signal. The main findings are summarized as follows:

- Using EOF analysis and outputs from OGCM experiments we find that the vertical density structure at both the western (WB) and eastern boundaries (EB) at 26N show characteristic signatures that can be unambiguously linked to buoyancy-forcing in the Labrador Sea.

- The vertical structure associated with the leading EOF mode of density variability on the WB (EOF1-WB) shows
positive anomalies from 1500 to 3000m depth that can be related to earlier changes in the North Atlantic deep water formation, and to density anomalies over the Labrador Sea, which are seen to lead PC1-WB by ~30 months. The PC1-WB is found to be very robust in both the CTRL and BUOY experiments, signalling buoyancy-forced AMOC variability on decadal timescales. PC1-WB explains 40% and 70% of the density variance for the CTRL and BUOY experiments respectively.

- The PC1-WB is found to lead density anomalies at 1000-1500m on the EB (associated with PC1-EB) by ~7months. The result of combining both boundaries into a single EOF allows to extract the correlated variance and the optimal





lag between the boundaries. The combined EOF variance shows maxima when lagging the EB between 7 months and 3 years. This lagged relationship is consistent with density propagation at 2700-3000m.


- In the CTRL experiment, density anomalies at 2000-3000m propagate southwards along the WB and eastward along the Equator and then up to the African coast impacting the vertical structure on both boundaries at 26N. The propagation is continuous from the Labrador Sea around the basin and up to the British Isles. This density signal propagates at a speed ~0.3m/s consistent with the propagation speeds in the BUOY experiment.


- The same method is applied to the RAPID array data for the common period with NEMO1 simulation. The two leading EOFs for the WB have anomalous densities at 1000m and 3000m respectively and are well simulated by the CTRL experiment (Fig. 8). This inter-annual variability is unequivocally wind-forced. The observational record has

to be longer in order to identify the buoyancy-forced vertical anomalies, which are more low-frequency signals.

- The same method was able to extract boundary signals from the higher resolution model HadGEM3-GC2. Despite the greater complexity in GC2, the vertical density profiles on the WB at 26N can be clearly related to the

geostrophic AMOC although some time-filtering is needed in order to separate the time-scales.

- After filtering (periods below and above of 2 years), PC1-WB is found to be more related to AMOC at 1000m (2000-3000m) at high (low) frequency with a EOF profile with positive density anomalies at 1000m and 3000m.

PC2-WB is also related to AMOC at 1000m at low frequency and shows positive (negative) density anomalies at deep 1000m (3000m) level.

- We also show clear density propagation from the Labrador Sea around the basin to the British Isles along a wave track (defined by 3138m bathymetry) at 3000m depth level, which it is also explaining part of the AMOC

variability. However, temporal filtering is needed to make this stand out above the noise.

- We conclude that the buoyancy-forced signal over the density profile at 26N will be captured in the observations (as well as in coupled models) if the available time period is long enough (>26 years), selecting density profiles with opposite anomalies at 1000m and 3000-3500m, with time filtering of periodicity >2 years, which would help

to eliminate high frequency wind-driven signals.

**Author contribution**

IP has made the calculations with the data from numerical simulations and observations. IP, JR and KH have analysed the results and IP prepared the manuscript and all authors have contribute to the writing and editing the manuscript.


**Competing interests**

The authors declare that they have no conflict of interest.

**Code and data availability**

The code used for the analysis is done in Matlab and it could be available by request to the first author. The data for the three forced experiments belongs to ECMWF and it could be available by request. GC2 data belongs to MetOffice and it could be available by request. RAPID data is fully available online at: http://www.rapid.ac.uk.

**Acknowledgments**


This work has been possible thanks to RAMOC project. I. Polo and C. Thomas have been funded through a NERC RAMOC grant. J. Robson is supported by the U.K. National Centre for Atmospheric Science-Climate (NCAS-Climate) via the ACSIS project, and K Haines is supported by NCEO and the University of Reading. We thank Magdalena Balmaseda for providing the NEMO experiments outputs and Martin Andrews and Pablo Ortega for providing the GC2 control data. Data from the
RAPID MOC monitoring project are funded by the Natural Environment Research Council and are freely available from www.rapid.ac.uk/rapidmoc. The observational program is part of the UK RAPID-AMOC program and full data policy is available online at: http://www.bodc.ac.uk/projects/uk/rapid/data_policy/.

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



**Table 1. Correlations between PCs f800m and AMOC timeseries in NEMO1**. Correlations between the geostrophic
AMOC and the PCs at the boundaries for the CTRL and BUOY experiments. Correlations are significant when are above 0.3
from a t-test at 95% confidence level.

| Correlation no filter/1yrm | PC1-WB AMOCg | PC1-EB AMOCg | PC1-EB-PC1-WB | |
|---|---|---|---|---|
| BUOY | 0.88/0.89 | 0.80/0.81 | 0.83/0.84 | |
| CTRL | 0.33/0.47 | -0.04/0.42 | 0.36/0.67 | |

**Table 2. Correlations between PCs f800m WB and AMOC timeseries in GC2.** Correlations between the geostrophic
AMOC and the PCs at the WB for the GC2 experiment. Correlations are significant when are above 0.2 from a t-test at 95%
confidence level.

| Correlation | Non filtered | < 2 years | >2 years | |
|---|---|---|---|---|
| PC1-WB, AMOCg | 0.45 | 0.49 | 0.25 | |
| PC2-WB, AMOCg | 0.13 | 0.06 | 0.32 | |
| PC1-WB, PC2-WB | 0 | 0.11 | 0.32 | |






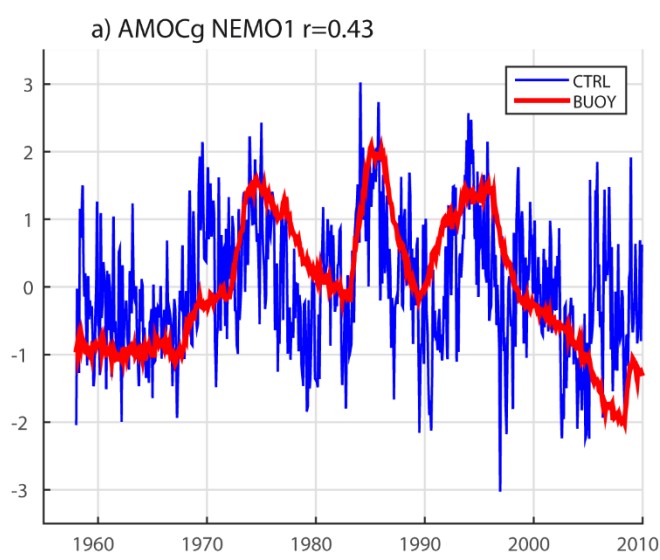

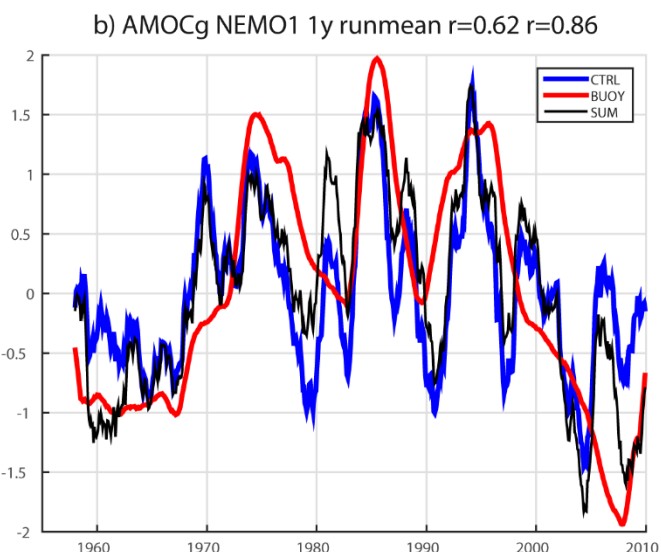

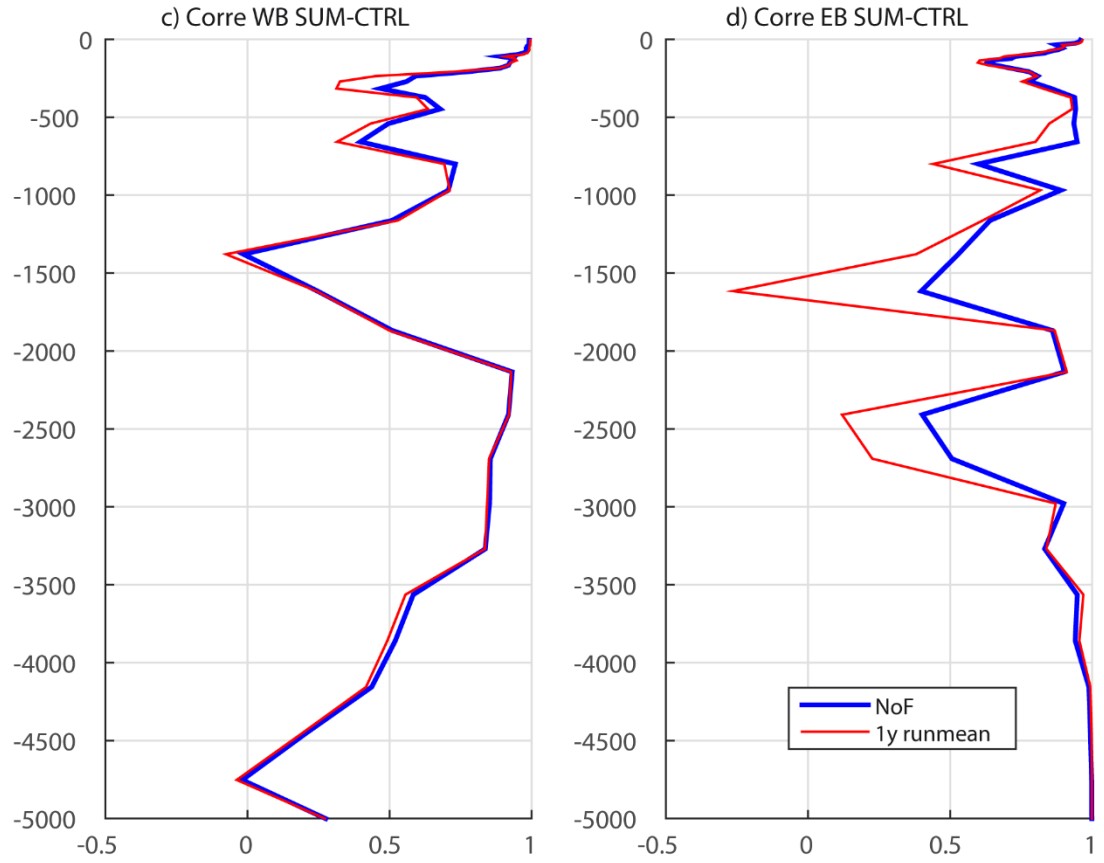





**Figure 1. AMOC and linearity of the forcings for the density profiles at the boundaries.** a) Geostrophic AMOC defined at 26N and

1100m level for the CTRL (blue line) and BUOY (red line) experiments. Timeseries have been de-seasonalised and de-trended and

standardised. Correlation in the title corresponds to correlation between blue and red lines. b) Same as a) but the timeseries have been

smoothed with one-year running mean filter and black line refers to AMOC for the SUM. In the title, double correlation score corresponds

to correlations between blue-red and blue-back lines. c) Correlation coefficient between profiles of density anomalies at the WB for the

CTRL experiment and SUM (BUOY+WIND). Blue line refers to anomalies without time filtering, red line refers to anomalies that have

been smoothed with one year running mean filter. d) Same as c) but for the EB.





**Figure 2 AMOC at 26N and individual EOF of density profiles at the 26N boundaries: sensitivity to depth truncation.** a) Timeseries associated with the leading mode of density variability of the water column at the WB-26N for the CTRL (blue line) and BUOY (red line) experiment, considering the total water column. Cyan line corresponds to the time series of the CTRL smoothed with one-year running mean filter. In the title, double correlation score corresponds to correlations between blue-red and blue-cyan lines b) Same as a) but for the EB-26N. c) -d) Same as a)-b) but considering from 800m downwards. All the timeseries are dimensionless and the percentage of explained variance is detailed on the legend as well as the correlation with AMOC timeseries (in Fig. 1a-b). The correlation between the CTRL and BUOY experiments timeseries are detailed in the title for the non-filter and the 1y running mean filtering. The EOFs are performed after removing seasonal cycle, linear trend and depth weighting pre-processing is applied.





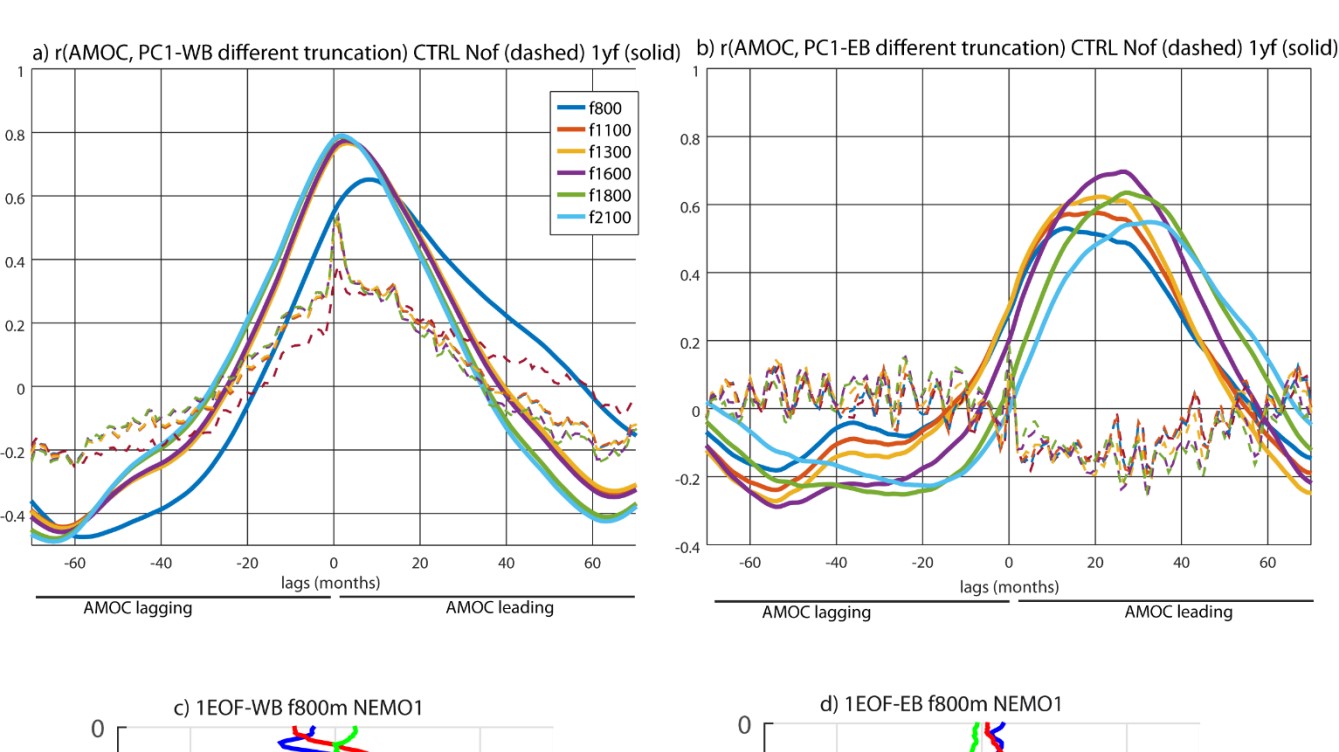

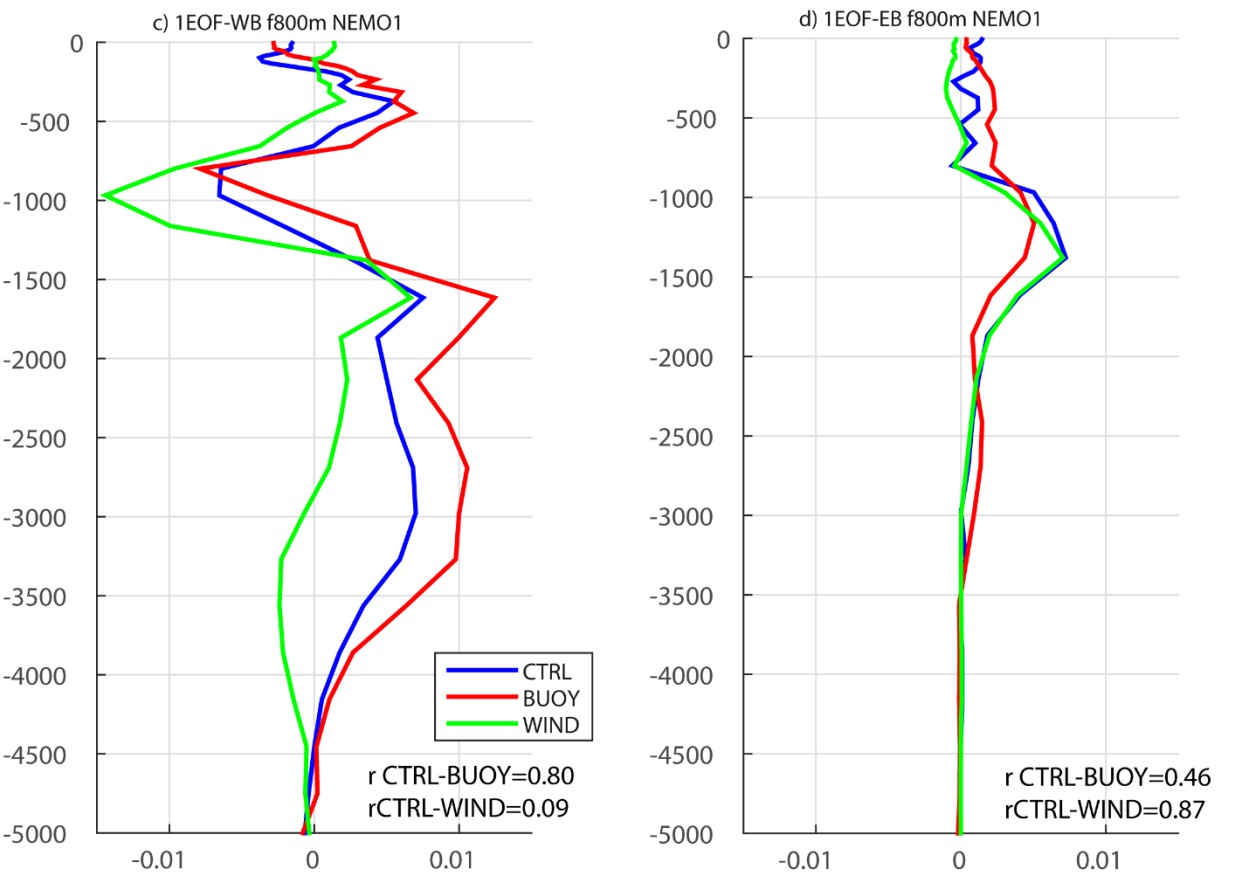





**Figure 3. Individual EOF of density profiles at the 26N boundaries from 800m.** a) Lead-lag correlations between AMOC at 1000m and 1PC-WB at different truncations (represented with different colors). Positive (negative) values over x-axis represents the AMOC leading
(lagging) the PC1-EB. 1-year running mean filter (solid line) and no filter (dashed line) is applied to the timeseries for the CTRL experiment. b) Same as a) but for the correlations between AMOC and PC1-EB. c) Profile of density anomalies (in kg/m³) associated with the leading mode of density variability of the water column from 800m downwards at WB-26N for the CTRL (blue line) and BUOY (red line) and WIND (green line) experiment. Associated timeseries are in Fig. 2c except for the WIND experiments. d) Same as c) but for the EB-26N, associated timeseries are in Fig. 2d except for the WIND experiment. Correlation coefficient between time series of the PCs for
CTRL with BUOY and WIND are detailed.





**Figure 4. Relationship between Boundaries and combined EOF**. a) Lead-lag correlations between PC1-WB from 800m and PC1-EB at different truncations (represented with different colors). Positive (negative) values over x-axis represents the PC1-WB leading (lagging)
the PC1-EB. 1-year running mean filter (solid line) and no filter (dashed line) is applied to the timeseries for the CTRL experiment. b) Same as a) but for the BUOY experiment and only 1-year running mean filter is plotted. c) - d) are the EOFs as the result of combining the density profile at the WB (from 800m) at lag 0 and the EB (from 1600m) at different lags. The CTRL experiment is plotted in blue and the cyan line corresponds to the EOF at the lag in which the explained fraction of variance is maximum (EOFmax). The latter equivalent EOFmax profile for the BUOY experiment is plotted in red. e) The associated timeseries of EOFmax for the CTRL (blue) and BUOY (red)
experiment.









**Figure 5. The spatial relationship of density anomalies with PC1-WB in the CTRL and BUOY experiment.** a) Density anomalies averaged from 2700m to 3000m levels, 30 months in advance projected onto PC1-WB (in kg/m³) for the CTRL experiment. Black lines correspond to the correlation every 0.2. Only significant areas are plotted with a Student's t-test with alpha=0.1 considering only effective degrees of freedom according to Metz (1991). b) and c) are same as a) but for lags -20 and -10 months. d)-f) same as a)-c) but for the BUOY experiment.



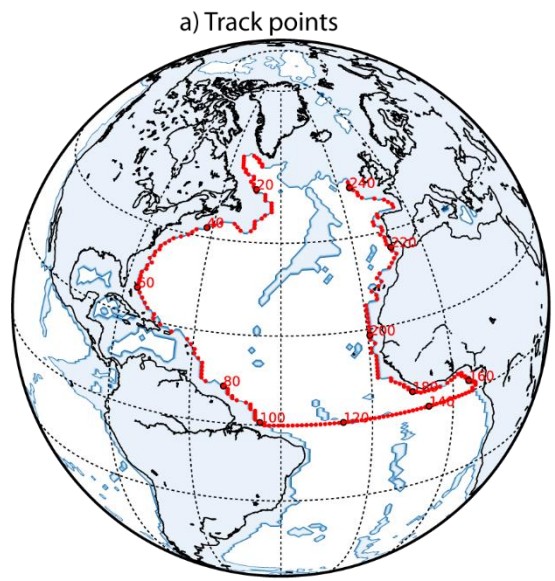

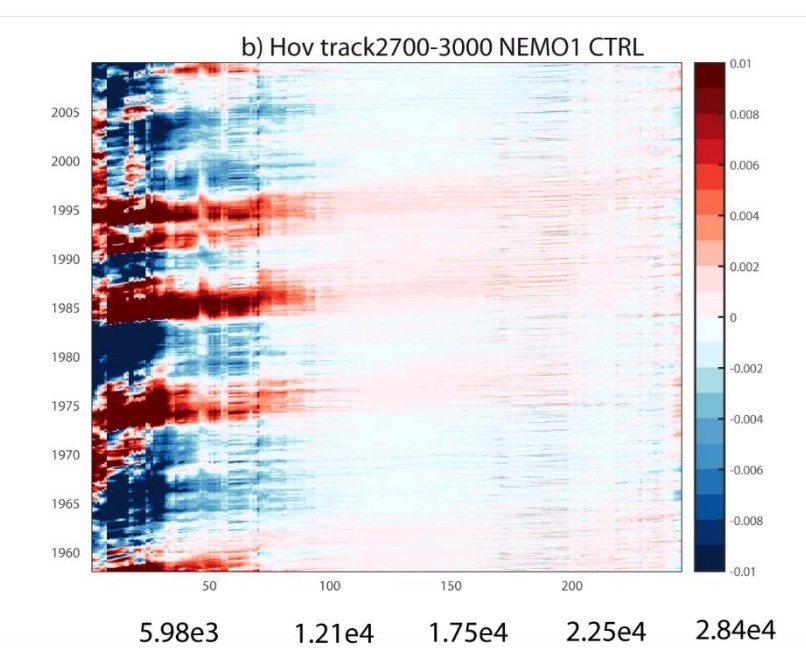

**Figure 6. Hovmöller diagrams of density anomalies along the wave-track in the CTRL experiment**. a) Map of the wave-track used in the study. The track-points correspond to the first point before the coast following the bathymetry at 3269 m. The points for all depth levels has been tracked. b) Density anomalies along the track averaged from 2700m to 3000m levels for the CTRL experiment (in kg/m³). The anomalies are the result of removing the linear trend and the seasonal cycle. Distance (in km) from Labrador Sea is associated with track-points in the x-axis.



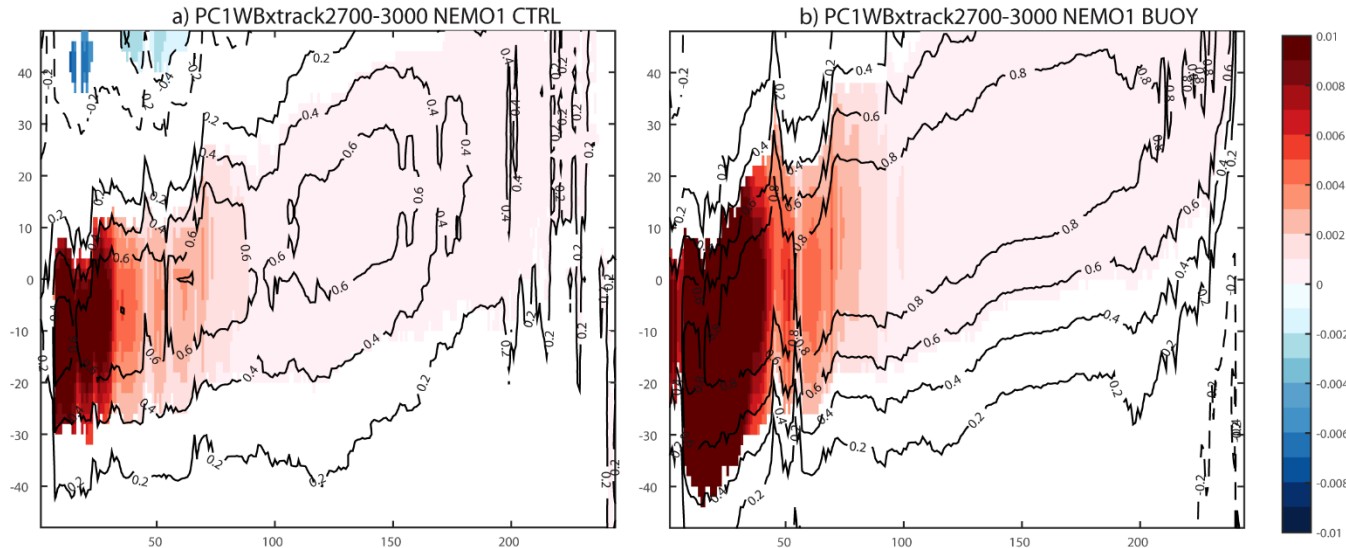

**Figure 7. PC1-WB and density along the wave-track**. a) Density anomalies averaged from 2700m to 3000m projected onto PC1-WB (in kg/m³) for the CTRL experiment. Black lines corresponds to the correlation every 0.2. Only significant areas are plotted with a Student's t-test with alpha=0.1 considering only effective degrees of freedom according to Metz (1990). b) Same as a) but for the BUOY experiment.


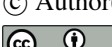


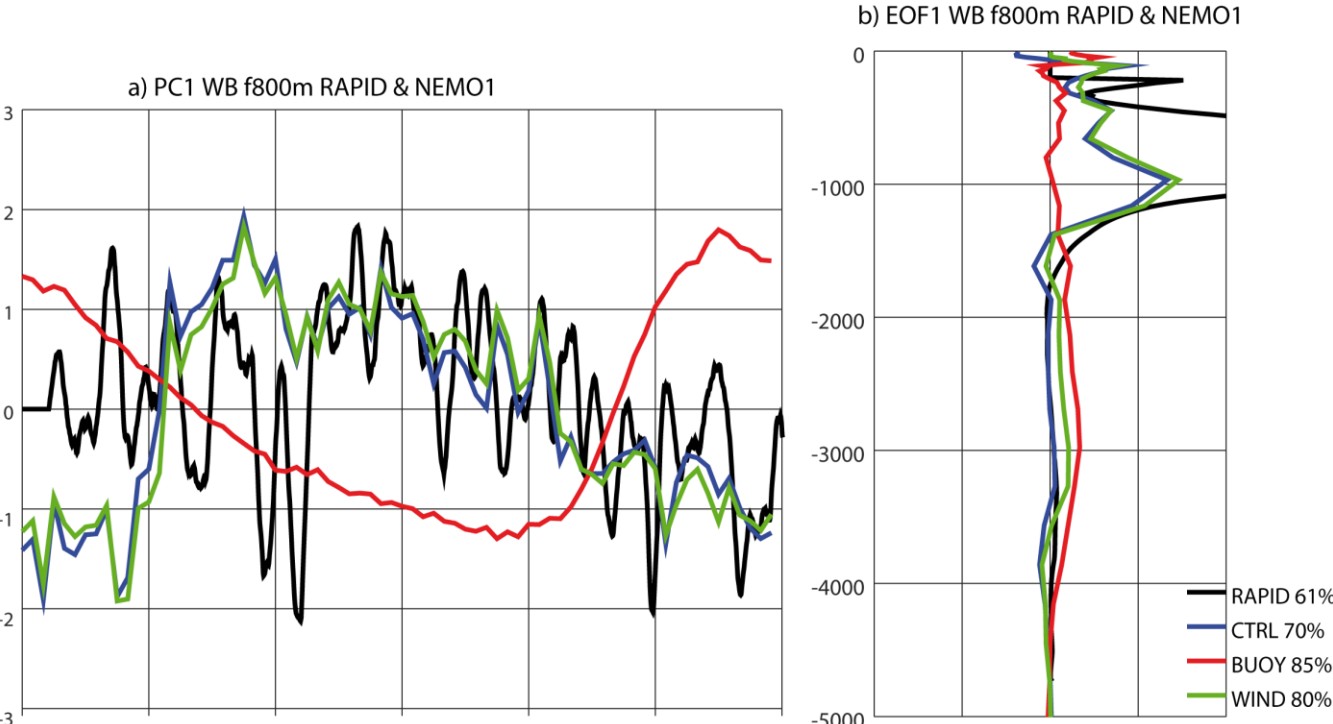

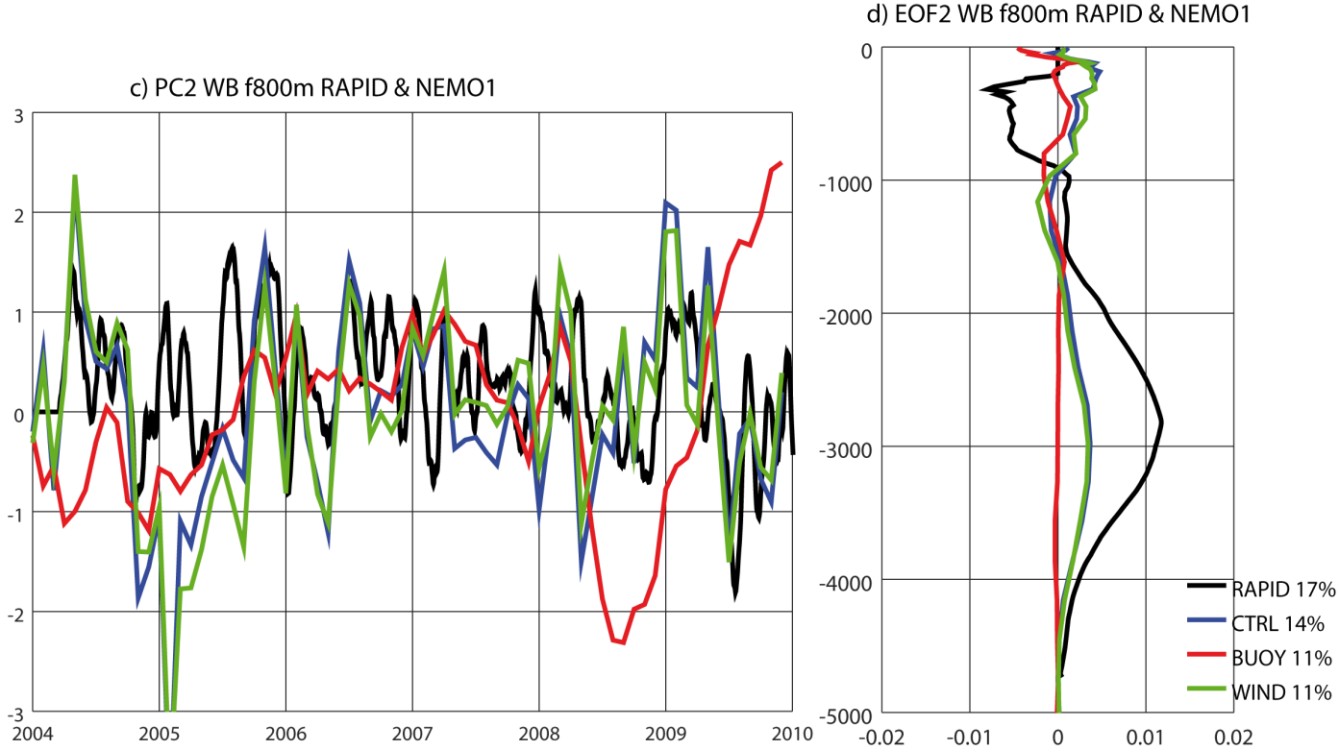





**Figure 8. Leading EOFs for the RAPID data and NEMO experiments**. a) Pcs Time series associated with the leading EOF for the WB-26N for the RAPID array (black line), the CTRL (blue line), WIND (green line) and BUOY (red line) experiments for the common period 2004-2010. b) EOF patterns linked to PC in a). c) -d) Same as a) -b) but for the second mode. For the period 2004-2010 the EOFs for the

CTRL experiment captures wind-forced only modes.



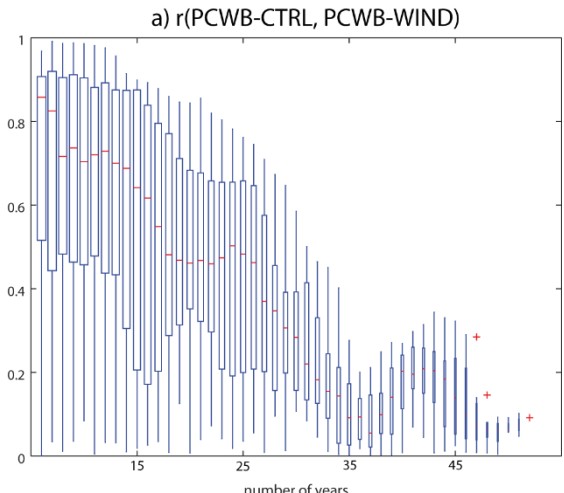

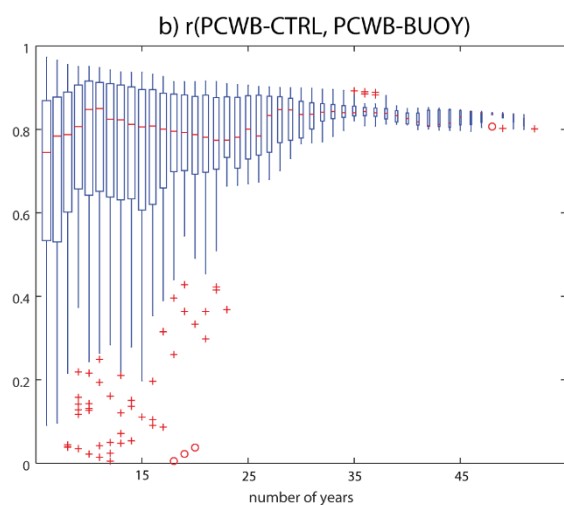

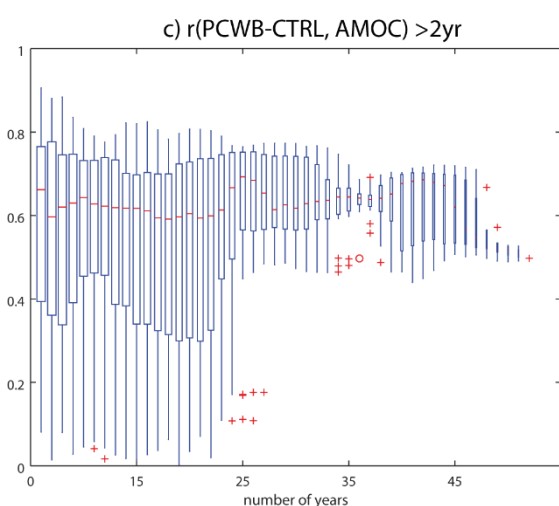





**Figure 9. Impact of the number of years used in capturing the buoyancy forced signal in CTRL.** a) box-plot of the correlation between leading PC-WB CTRL experiment and the leading PC-WB WIND experiment for different extension of the time-periods. b) Same as a) but for the correlation between PC1-WB CTRL and PC1-WB BUOY. c) Same as a) but for the corrrelation between PC1-WB CTRL and AMOC filtered with periods >2 years. The anomalies have been calculated for the long-term mean and detrended over the whole period prior the PC analysis for all experiments. Red line corresponds to the median, points between 1.5 and 3 times the Inter-Quartile Range (IQR) is marked with crosses and and points outside 3 times IQR with circles. Y-axis corresponds to correlation between PCs, x-axis corresponds to number of years in all sub-periods considered. The signal extracted by 1PC-WB CTRL is always a mix of forcings however, high probability of extract buoyancy-forced signals when periods are longer than ~26 years. If period are longer than 35 years, the wind-forced signal is negligible.




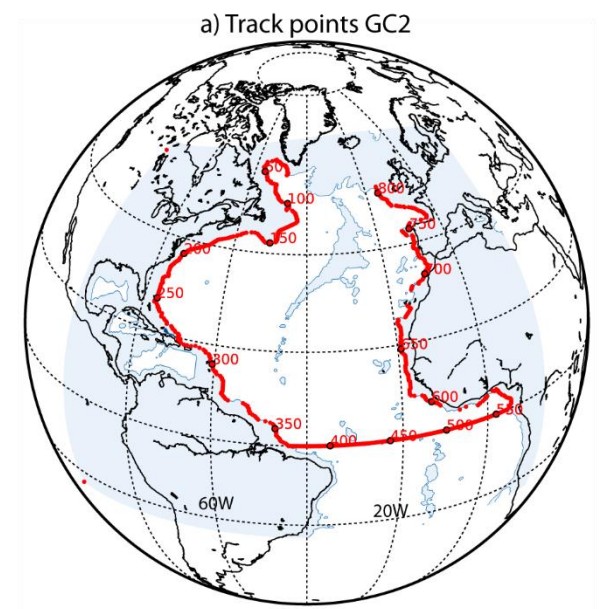

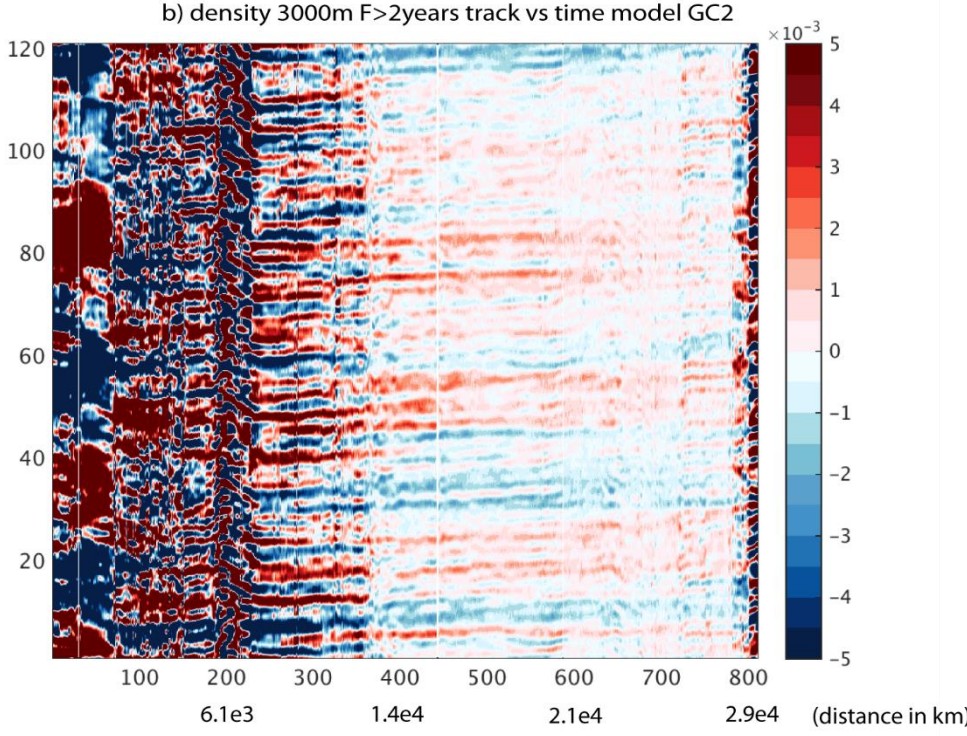

**Figure 10. Hovmöller diagram of density along the wave track in the GC2 experiment.** a) The track-points correspond to the first
point before the coast following the bathymetry at 3138 m in GC2. b) Hovmöller diagram of density anomalies at 3000m along the wave





track, after filtering the density timeseries retaining >2 years. X-axis corresponds to track points in Fig. 10a and y-axis corresponds to model year.









**Figure 11. EOFs of WB density profiles at 26N in the GC2 experiment.** a) Leading (blue line) and second (red line) EOF from WB at 26N and 76.75W in the GC2 experiment. Percentage of explained variance and correlation with the geostrophic AMOC timeseries defined at 26N and 925m depth are displayed. b) Timeseries associated with leading EOF-WB (PC1-WB, blue line), PC1-WB filtered timeseries (PC1-F>2y, brown line) and the AMOC filtered timeseries (AMOC-F>2y, green line). c) Same as b) but for the PC2-WB. d) Correlation between AMOC (at lag 0) and the first two PCs-WB (lagged). e) Correlation between PC1-WB and AMOC-Ek at 900m and 26N. Box-plot

similar than Fig. 9c, x-axis represents number of years used in the periods for the correlation, y-axis represents correlation score. f) Same as e) but for the correlation between PC2-WB and AMOC-Ek at 900m and 26 and filtered with periodicity > 2 years.