# Peer review of "Can the boundary profiles at 26N be used to extract buoyancy-forced Atlantic Meridional Overturning Circulation signals?"

_Ocean Science, 2020_

## Referee Comment (RC1) · Anonymous Referee #1 · 22 Apr 2020

This paper addresses a significant question and presents interesting results which should be published. There are though some important details that need to be addressed in the presentation of the results before the work can be published.

The main result is that, in the model simulations, the variance of density variations on the boundary is, like the AMOC, dominated by the wind on inter annual timescales and only on decadal timescales does the buoyancy forced signal emerge from the "noise" of the wind forced variability. This is not too surprising and the paper could be strengthened by more emphasis on the structure of the variations associated with wind and buoyancy forcing, which has been less studied in previous work.

Below are some specific points to consider:

1. More often than not the axes in figures are not labelled and/or units are not given. This is a fundamental requirement.

2. In the introduction (l95) it is staled that section 7 "describes the limitations of the interpretations, as well as possible applications e.g. in data assimilation", but there is no mention of assimilation in section 7.

3. Table 1. Its stated that all correlations are significant if greater than 0.3, but doesn't the significance of correlation depend upon number of independent data and hence whether the time series ae filtered or not?

4. Figure 1b. I would like to see the time series of WIND shown too, without that it is not clear how much variability there is in wind on longer timescales.

5. Figure 1a,b Was the same standardisation applied to each time series? Probably better not to standardise then it would be obvious to the reader what is being compared.

6. Notation in the figures is sometimes inconsistent. E.g. "AMOCg" and "1PC" in Figure 1 , but "AMOC" and "PC1" elsewhere. Also abbreviations identifying filtered and unfiltered data vary.

7. Calculation of EOFs (l228) Figure 3c,d show EOFs calculated using data below 800m, so how are these extended up to the surface as seen in the Figures? Some more information is needed in the methods. And are these EOFs for annual or monthly data?

8. The EOFs presented in this paper are dimensional. This is useful, but they are not always calculated this way (sometimes they are normalised) so the text need to make clear what methodology is used here. And it is particularly important to show units on axes to avoid confusion! I presume the variance of the PCs is standardised?

9. The results of different truncations in Figure 4 suggest anomalies propagate at different speeds at different depths. This would imply that there is no direct correspondence between the EOFs and the modes of boundary waves. This should be discussed.

10. Discussion of Figure 8 should highlight how low the variability is in run BUOY over this period.

11. To what extent does the length of the model run affect the results in Figure 9? The run is only about 60 year long so I suspect that for the longer time periods the reduction in the inter-quartile range is a reflection of the length of the model run, rather than the uncertainty of the correlation.

Other minor comments:

l15 "a propagation speed leading to" is unnecessary

l16 Abstract - what does maxima refer to?

l67 Despite many model studies (insert 'model')

Fig 9 "extension of the " is not needed. Often the captions are not as well written as the text.

l261 More variance of what?

l459 What does "simultaneously" refer to?

l460 "It is worth notice that " Also rephrase to making meaning clear " . . . that, due to . . . , time filtering is . . ."

l470 "but it is not that obvious for the EB" not clear what this is referring to.

l476 Which PC1 NEMO (Control, WIND or BUOY? and evaluated over what time period?

l490 1st conclusion - expand what "characteristic signatures " means

---

## Referee Comment (RC2) · Anonymous Referee #2 · 23 Apr 2020

In manuscript "Can the boundary profiles at 26N be used to extract buoyancy-forced AMOC signals?", Polo et al. examined vertical profile of density in at 26N and its relation to AMOC and to the buoyancy forcing subpolar North Atlantic, in both forced and couple models. Their results suggest that depth structure and the lagged covariances between west and east boundaries at 26N may provide useful information for detecting density anomalies of high latitude origin in more complex model and observations, although time filtering and longer time series are required. The paper is well-organized and well-written over all, but I do have a couple of concerns regarding the realism of the 1 degree model and how fast the density signal transferring from subpolar to subtropical North Atlantic and some clarification and/or discussion are needed.

Title: Spell out AMOC

[Figure]

Line 7: "The temporal variability of the Atlantic meridional overturning circulation (AMOC) is driven ..."

L135: Given the central relevance to the topic, it is required to include a simple comparison of the T/S/density profiles between model and RAPID for, let's say, a five-year period 2005-2009, to see characteristically how similar/difference they are. And this should be in the paper, not in the supplementary, I understand that profile of density gradient is shown in supplementary, which is fine. Why focus on density gradient there?

L149: Here I think the modeled standard deviation for monthly mean AMOC need to be listed too for comparison with the RAPID value (4.4 Sv).

L215: Should Fig. 1a be Fig. 1c?

L299: The speeds in the forced models are consistent with the lag found between boundaries, but more importantly, are these speeds realistic? It needs some discussions regarding how fast the density signal transfers. A speed of 0.3-0.4 m/s seems really fast to me (until I saw 2m/s later in GC2 experiment), what does this speed represent? Do we have observational/theoretical supports? The subpolar density exhibits a significant variability in the last several decades, why we do not see similar variability in the south.

L314 and later in this section: 2004-2010 should be 2004-2009

L325: It seems to me an overstatement that the model and observations are very similar here. Not only the PC differ significantly for the first half of the time series, but also the model EOF pattern exhibit much lower magnitude. Also, is the length of the record the only key factor that leads to the difference between BUOY and CTRL (and RAPID)? Although Figure 1b with longer record does show a significant correlation between CTRL and BUOY (in AMOC), the similarity is mainly due to the decadal variability during 1970s-1990s, for which I am not sure if there is any observational support. The similarity/difference between the two experiments display significant time-dependence.

L372: The phase speed in GC2 is faster than in the forced simulations by a factor of 5-7 and clearly need some explanations. How robust are the model results, especially, to different model resolution?

---

## Author Comment (AC1) · 18 Jun 2020

Anonymous Referee #1 This paper addresses a significant question and presents interesting results which should be published. There are though some important details that need to be addressed in the presentation of the results before the work can be published. The main result is that, in the model simulations, the variance of density variations on the boundary is, like the AMOC, dominated by the wind on inter annual timescales and only on decadal timescales does the buoyancy forced signal emerge from the "noise" of the wind forced variability. This is not too surprising and the paper could be strengthened by more emphasis on the structure of the variations associated

with wind and buoyancy forcing, which has been less studied in previous work. Below are some specific points to consider:

1. More often than not the axes in figures are not labelled and/or units are not given. This is a fundamental requirement.

Response: Thank you for the comment. We have carefully checked the figures and the plots are now correctly labelled with units.

2. In the introduction (l95) it is stated that section 7 "describes the limitations of the interpretations, as well as possible applications e.g. in data assimilation", but there is no mention of assimilation in section 7.

Response: We have removed the sentence from the introduction.

3. Table 1. Its stated that all correlations are significant if greater than 0.3, but doesn't the significance of correlation depend upon number of independent data and hence whether the time series are filtered or not?

Response: Thank you for this comment. We have calculated the significance with a t-test now with the number degrees of freedom. In table 1 and 2, we have highlighted the significant correlations with a star. The full table of p-values with corresponding effective number degrees of freedom for a two-sided t-test is detailed for the referee below:

p-value (number of effective degrees of freedom) Nf / 1yrF r(PC1-WB, AMOC) r(PC1-EB, AMOC) r(PC1-EB,PC1-WB) BUOY 0.63/0.63 (10/10) 0.75/0.75 (7/7) 0.67/0.67 (9/9) CTRL 0.2/0.38 (103/26) 0.20/0.37(241/29) 0.25/0.48 (64/17)

4. Figure 1b. I would like to see the time series of WIND shown too, without that it is not clear how much variability there is in wind on longer timescales.

Response: We have added the AMOC timeseries for the WIND experiment in new Fig. 2, to compare variability at long timescales, filtered with 1year running mean. The

WIND experiments show variability with a peak in 5 years-period in comparison with BUOY and CTRL which has 10 years-period.

5. Figure 1a,b Was the same standardisation applied to each time series? Probably better not to standardise then it would be obvious to the reader what is being compared.

Response: Yes, the timeseries are standardised (for each experiment with respect to their own seasonal standard deviation), in order to compare the variability. Otherwise the comparison could be not fair due to, in part, the different model drifts in each experiment (Megann et al 2014). We have added panels to the Fig. S2 for this clarification. In Fig. S2a it can be appreciated that BUOY experiment has a strongly positive trend compared with CTRL and WIND. Fig. S2b shows the seasonal cycle of the interannual variability. The standard deviation (in Sv) for the 3 experiments is very different; In particularly there is a strong seasonal dependence for CTRL and WIND, but relatively little change through the year for the BUOY experiment. Therefore, we think it is better to compare the normalised anomalies.

6. Notation in the figures is sometimes inconsistent. E.g. "AMOCg" and "1PC" in Figure 1 , but "AMOC" and "PC1" elsewhere. Also abbreviations identifying filtered and unfiltered data vary.

Response: AMOC and PC1 are now consistently used through the manuscript, and in the figures. We also use Nf (for no filtered) and 1yrF (for 1year filtered).

7. Calculation of EOFs (l228) Figure 3c,d show EOFs calculated using data below 800m, so how are these extended up to the surface as seen in the Figures? Some more information is needed in the methods. And are these EOFs for annual or monthly data? 8. The EOFs presented in this paper are dimensional. This is useful, but they are not always calculated this way (sometimes they are normalised) so the text need to make clear what methodology is used here. And it is particularly important to show units on axes to avoid confusion! I presume the variance of the PCs is standardised?

Response: We have better clarified now in the methods Section how PCs are calculated (from line 172). PCs are calculated from monthly density data. The PCs for below 800m (PC-f800) are calculated using only the information from 800m downwards. The PC corresponds to standardised anomalies. However, the EOF pattern is represented as the regression of density anomalies onto the PC-f800. Therefore, the resulting EOFs can have a density signal shallower than 800m. Furthermore, EOFs have density units (density anomalies per 1 standard deviation of the PC).

9. The results of different truncations in Figure 4 suggest anomalies propagate at different speeds at different depths. This would imply that there is no direct correspondence between the EOFs and the modes of boundary waves. This should be discussed.

Response: Thank you for the comment. The correspondence of PCs and boundary wave modes is not direct.

The boundary EOFs are not stationary with time and latitude. Figure 4 shows that the teleconnection between density signals at WB and EB occurs very quickly at 800m-1100m (around 8 months as a fast response) however, at deeper levels for EB, the signal is delayed. There is a possible vertical propagation of the density anomaly at the WB from upper levels (400m) to 2700m.

Different lags in old Fig. 4 could be reflecting different propagation speeds along the equator (Hovmuller diagrams at different depths suggest that the phase speed changes at the equator, not shown here). Whereas the propagation along either the WB or the EB are reasonable to represent similar boundary waves, with coherent combined EOF vertical mode, albeit different modes on WB and EB.

10. Discussion of Figure 8 should highlight how low the variability is in run BUOY over this period.

Response: the AMOC in the BUOY experiment has a peak variability at 10 years (figure 1 in Polo et al., 2014). Similarly, the variability of the zonal density gradient also has a

very low frequency in the BUOY experiment (figure 2 in Polo et al 2014). Additionally, it is shown that EB density variability contains higher frequencies in comparison with WB density variability in BUOY (see new Fig. 3c, d).

Considering the short period analysed in Fig. 8 (2004-2009), the BUOY experiment shows a decrease and increase of AMOC at 26N (it is also seen in new Fig. 2 at the end of the period). This trends in AMOC are associated with density variability at WB (red line in new Fig. 3c). Therefore, we agree with the referee about the WB low-frequency density signals in BUOY experiment for long (1958-2009) and short (2004-2009) periods.

However, RAPID shows a decline from 2007 up to 2010, more similar to CTRL and WIND timeseries (new Fig. 9). As a conclusion, RAPID at 2004-2009 period is only representing wind-forced signal. We have tried to make it clear in Section 5 (from line 334).

11. To what extent does the length of the model run affect the results in Figure 9? The run is only about 60 year long so I suspect that for the longer time periods the reduction in the inter-quartile range is a reflection of the length of the model run, rather than the uncertainty of the correlation.

Response: The referee is right. As the period is longer with more years, we have fewer independent periods and therefore less variance in the sample distribution (i.e. the box-plot is smaller). However, the interpretation of old Fig. 9 is still valid. Motivated by old Fig. 8, we realized that for short periods, it was difficult to discriminate buoyancy-forced signals in CTRL (and RAPID). Old Fig. 9 extends this idea using samples of the total model period. Old Fig. 9 (new Fig. 10) suggests that for long periods the correspondence of PC1-WB CTRL and PC1-WB BUOY is straightforward (correlation is high and thus the WB density profile is mainly buoyancy-forced signal).

Other minor comments: l15 "a propagation speed leading to" is unnecessary

Response: We have removed that part in the sentence.

l16 Abstract - what does maxima refer to?

Response: The sentence has been changed to "The timing of the density anomalies appearing at the eastern and western boundaries at 26N reveals ∼2-3 years lags between boundaries at the deeper levels (2600-3000m)". (line 15)

l67 Despite many model studies (insert 'model')

Response: We have inserted the word in the sentence.

Fig 9 "extension of the " is not needed. Often the captions are not as well written as the text.

Response: We have rewritten the caption for figure 9 and also we have checked all the figure captions.

l261 More variance of what?

Response: We have rewritten the sentence to be clearer. The first EOF explains part of the density variance.

l459 What does "simultaneously" refer to?

Response: We have removed the word in the sentence.

l460 "It is worth notice that " Also rephrase to making meaning clear " : : : that, due to : : : , time filtering is : : :"

Response: The sentence has been rephrased as follows: "It is worth noticing that, due to noise from high frequency wind-forcing, time filtering is needed to see a clear signal in GC2."

l470 "but it is not that obvious for the EB" not clear what this is referring to.

Response: The sentence has been rephrased as follows: ". . . but the propagation up

the EB is less clear.".

l476 Which PC1 NEMO (Control, WIND or BUOY? and evaluated over what time period?

Response: "The temporal variability in RAPID WB density modes suggests that wind forcing is still dominating at these short timescales."

l490 1st conclusion - expand what "characteristic signatures " means.

Response: The first conclusion is that we have found density vertical profiles linked to AMOC. Then we explained the characteristics in the following conclusions.

---

## Author Comment (AC2) · 18 Jun 2020

Anonymous Referee #2 In manuscript "Can the boundary profiles at 26N be used to extract buoyancy-forced AMOC signals?", Polo et al. examined vertical profile of density in at 26N and its relation to AMOC and to the buoyancy forcing subpolar North Atlantic, in both forced and couple models. Their results suggest that depth structure and the lagged covariances between west and east boundaries at 26N may provide useful information for detecting density anomalies of high latitude origin in more complex model and observations, although time filtering and longer time series are required. The paper is well-organized and well-written over all, but I do have a couple of concerns regarding the realism of the 1 degree model and how fast the density signal transferring from subpolar to subtropical North Atlantic and some clarification and/or discussion are needed.

Title: Spell out AMOC.

Response: The title has been changed.

Line 7: "The temporal variability of the Atlantic meridional overturning circulation (AMOC) is driven : : :"

Response: The sentence has been changed

L135: Given the central relevance to the topic, it is required to include a simple comparison of the T/S/density profiles between model and RAPID for, let's say, a five-year period 2005-2009, to see characteristically how similar/difference they are. And this should be in the paper, not in the supplementary, I understand that profile of density gradient is shown in supplementary, which is fine. Why focus on density gradient there?

Response: As the referee has suggested, we have added the new Fig. 1, which shows the comparison of the temperature, salinity and density profiles at the boundaries between RAPID and CTRL experiments for the period 2005-2009. The figure is commented in the manuscript (from line 138).

Vertical density gradients could be important for the boundary waves phase speed, so we will maintain Fig. S1, which also shows the standard deviation of the density profiles.

L149: Here I think the modeled standard deviation for monthly mean AMOC need to be listed too for comparison with the RAPID value (4.4 Sv).

Response: The standard deviation, after removing the linear trend, using the monthly mean timeseries for NEMO1 is 2.28 and 1.92 Sv for CTRL and BUOY respectively. RAPID reaches double this level (4.4 Sv). This information is now added (from line

153). Additionally, the seasonality of the standard deviation is shown in Fig. S2.

L215: Should Fig. 1a be Fig. 1c?

Response: The reference to Fig. 1a was right. The correlation scores in old Fig. 2c are the correlations between timeseries in old Fig. 2c and timeseries in old Fig. 1a.

L299: The speeds in the forced models are consistent with the lag found between boundaries, but more importantly, are these speeds realistic? It needs some discussions regarding how fast the density signal transfers. A speed of 0.3-0.4 m/s seems really fast to me (until I saw 2m/s later in GC2 experiment), what does this speed represent? Do we have observational/theoretical supports?

Response: There are works showing the density propagation along western boundary (at deeper levels) from variations of the AMOC (i.e. Kohl, 2005; Hodson and Sutton, 2012; Zhang, 2010). Hodson and Sutton (2012), although they show spatial patterns of the adjustment, they do not specifically give a phase speed of boundary propagation. However, they showed how the speed of the adjustment is sensitive to model resolution, they found that higher resolution model represents better the propagations at tropical latitudes. Zhang (2010) found different regimes along the latitudinal wave path. From 34N to the equator, she found a propagation coherent with Kelvin wave phase speed.

Johnson and Marshall (2002) have theorised the surface response to an AMOC change through the wave propagation. Other studies have found Kelvin wave propagation from observations and models in the equatorial Atlantic. From linear theory, Atlantic equatorial Kelvin wave speed for first, second and third baroclinic modes are around 2.5, 1.3 and 0.8 m/s respectively (Katz, 1997; França et al., 2003; Illig et al., 2004; Polo et al., 2008). In particular, from sea surface height, Polo et al., (2008) found Kelvin-like waves along the equatorial Atlantic and poleward along the African coast traveling at 1.6m/s, as mixed first and second baroclinic modes. We had reviewed the adjustment of the AMOC on different model simulation in the introduction section 1, and now we

have added more in section 4 (from line 390).

The subpolar density exhibits a significant variability in the last several decades, why we do not see similar variability in the south.

Response: Density over the subpolar gyre is highly variable in comparison with lower latitudes. We assume that part of the energy is lost as it travels southward. The Labrador Sea is a major source of density variations on the western boundary at depth in many models (Hodson and Sutton, 2012, Polo et al 2014). The propagation in figure hovmuller is quasi-non-dispersive until the equator (around track point 100).

L314 and later in this section: 2004-2010 should be 2004-2009

Response: it is now corrected.

L325: It seems to me an overstatement that the model and observations are very similar here. Not only the PC differ significantly for the first half of the time series, but also the model EOF pattern exhibit much lower magnitude. Also, is the length of the record the only key factor that leads to the difference between BUOY and CTRL (and RAPID)? Although Figure 1b with longer record does show a significant correlation between CTRL and BUOY (in AMOC), the similarity is mainly due to the decadal variability during 1970s-1990s, for which I am not sure if there is any observational support. The similarity/difference between the two experiments display significant time-dependence.

Response: In general, variability in the observations is greater than in the model (see previous comparison of AMOC standard deviations). This is also true in density profiles (see Fig. 1 and Fig. S1 RAPID has more variance at both boundaries at nearly all levels. This is especially important for NEMO1, the low-resolution model does not resolve part of the density variability.

Decadal variability in the AMOC is also supported by the reanalysis products (Wang et al 2010). The similarity of CTRL and BUOY operates at lower frequencies (decadal) not at interannual, which is dominated by the wind (see also Polo et al 2014). This is

also true for the comparison we made with RAPID, with only a short record we cannot extract buoyancy-forced signal from the CTRL experiment (new Fig. 9). We have made an effort to explain better this part in the manuscript (from line 332).

L372: The phase speed in GC2 is faster than in the forced simulations by a factor of 5-7 and clearly need some explanations. How robust are the model results, especially, to different model resolution?

Response: There are factors that potentially can influence propagation (and hence the phase speed) of boundary waves in numerical models. i) Model resolution; the along-shore phase speed of a Kelvin wave falls rapidly as grid spacing increases beyond the Rossby radius (Hsieh et al (1983)). Using 2 coupled models, with Arakawa-B grid, Hodson and Sutton (2012) showed that the MOC adjustment proceeds more rapidly in the higher resolution model due the increased speed of western boundary waves. Although in our study, NEMO1 model uses an Arakawa-C grid, it is expected to show difference in higher resolutions versions. ii) Lateral viscosity; increased values of viscosity reduce the along-shore phase speed of coastal Kelvin waves (Davey et al., 1983)). iii) Orientation of the coastal boundary relative to the ocean grid: the along-shore Kelvin wave speed falls as the angle of the coastline to the underlying grid increases (Schwab and Beletsky, 1998). iv) ocean stratification along the path wave.

Additionally, in our study, GC2 is a more complex model since it is a coupled model in comparison with ocean-only NEMO1, which potentially influence the propagation. Hodson and Sutton (2012) found in coupled model evidence of extratropical atmosphere response and a weak negative feedback on deep density anomalies in the Labrador Sea. We think that both, resolution and complexity are explaining the differences in the phase speed along boundaries shown in the hovmuller plots (new Figures 7 and 11). However, specific sensitive experiments should be designed and performed in order to properly understand these factors. This is now discussed more in section 7 (from line 480).

---

## Author Response (AR2)

Comments to the Author:
Please carefully consider the reviewer's comments below. Previous comments were not addressed satisfactorily. Clearly there are many minor details which should be improved. The manuscript requires a careful proofreading (beyond the few points picked up by the reviewer). Please also carefully edit the figures for labels, units and presentation (which is of fair quality and not acceptable for publication).

Response: Thank you for your comments, we have made a substantial effort to correct the minor mistakes as well as the English grammar. We have also checked all labels and units on the figures, see some particular issues addressed below.

Referee#

In the authors' response to the reviewers' comments it would have been helpful if they stated explicitly how the paper has been changed in response to the comments. For example, in response to comments 9 and 11 the authors discuss the comment made by the review, but only by reading the paper is it clear that no changes were made, Mostly the authors have responded but there are still details of the paper that should be improved.

Response: Regarding comment 9 *("The results of different truncations in Figure 4 suggest anomalies propagate at different speeds at different depths. This would imply that there is no direct correspondence between the EOFs and the modes of boundary waves. This should be discussed")*, we have added in the discussion section a paragraph (line 505) about the PCs and boundary wave modes.

*"We want to comment that the correspondence of the PCs and boundary wave modes is not direct. The boundary EOFs only capture variability at single locations (26N). Figure 4 shows that the teleconnection between density signals at the WB and EB occurs very quickly at 800m-1100m (around 8 months as a fast response) however, at deeper levels the EB signal is delayed. The different lags in Fig. 5 reflect different propagation speeds at different depths, mainly along the equator (Hovmöller diagrams at different depths suggest that the phase speeds change at the equator). The propagation along either the WB or the EB represent boundary waves with more coherent EOF vertical modes, albeit different modes on the WB and EB."*

Regarding comment 11 *("To what extent does the length of the model run affect the results in Figure 9? The run is only about 60 year long so I suspect that for the longer time periods the reduction in the inter-quartile range is a reflection of the length of the model run, rather than the uncertainty of the correlation)."*

It is true that the dispersion (inter-quartile range) is reduced because we have less sub-periods in the sample, but we note that the median is also dramatically changed with the number of years (figure 10a). The interpretation of figure 9 needs to consider both factors.

We have added in section 5 (line 357) a sentence highlighting the difference between boxes in Figure 10.

*"We note that as the sub-periods get longer, with more years (x-axis), we have fewer independent sub-periods and therefore less dispersion in the sample distribution (i.e. the box is smaller). However Fig. 10 still suggests that for long periods the correlation between PC1-WB CTRL and PC1-WB BUOY is high, correlation between PC1-WB CTRL and PC1-WB WIND is low, and thus the WB density profile associated with PC1-CTRL is mainly buoyancy-forced signal."*

We have also added in the caption of Figure 10 the following sentence:

*"We have to note that, for sub-periods with more years we have a smaller number of independent sub-periods and therefore there is less dispersion in the sample distribution (i.e. the box is smaller). "*

Axes are now labelled (Comment 1 in initial review). But there are still errors in the labels e.g. lower case 'k' should be used in 'km' and 'kg', and the units of temperature are °C not C

Response: These errors have been corrected

The captions still need to be proof read. E.g. Figure 2 "Timeseries have been de-seasonalised and de-trended and standardised." The first "and" should be a comma. Figure 3 "The EOFs are performed after removing seasonal cycle, linear trend and depth weighting pre-processing is applied." is not correct, but maybe it is not needed at all as the details are in the text. These are examples and I have not checked the paper thoroughly. I recommend that all the authors proof read all of the paper

Response: These figure captions have been corrected.

My original comment 10 was not well expressed. What I meant was that a change in density of 0.001 kg/m^3 is extremely small and would be very difficult to detect in observations.

Response: The referee is right; the anomalies of BUOY experiment are very small (in EOFs in Figure 9). However, the conclusion of Figure 9 is precisely that the BUOY signal for these short periods is not easily extracted, while the observed EOFs are mainly representing wind forced signal.  We have added a sentence highlighting the small amplitudes for the BUOY experiment (line 335):

*"However in BUOY (Fig. 9, red lines), although the deep 3000m peak shows up as EOF1, it has very small amplitude (0.001 kg/m3) showing very little variability over the short 2004-2009 period."*

About comment 11 and the new figure 10. While it may not affect the conclusions, I do think that it is important to highlight the impact of the length of the period on the interquartile range of the correlations. Not all box plots are the same so I think the caption to Figure 10 should state what the boxes and lines represent. Presumably "points between 1.5 and 3 times the Inter-Quartile Range (IQR) are marked with crosses" should be "points between 1.5 and 3 times the Inter-Quartile Range (IQR) from the median"

Response: We have revised the caption of Figure 10 as requested. We have also answered the points about time periods in the previous comment, and we have added a few more sentences about the box-plots in the main text (line 357).

[revised manuscript text omitted]

---

## Author Response (AR3)

Comments to the Author:

Thank you for addressing the minor comments. I request you to do one more minor technical correction before the manuscript is accepted for publication.
PSU is not a recognized salinity unit. You can call it practical salinity, or salinity measured in practical salinity scale, and this does not have a unit.
Please correct the salinity in line 139 (of the marked-up version) as practical salinity or salinity measured in practical salinity scale.
Please remove (psu) from x-label of Fig 1 and correct the caption (remove psu).

Response: We have done all the changes required by the Editor. Thank you for your comments.

[revised manuscript text omitted]